# ΠNET: OPTIMIZING HARD-CONSTRAINED NEURAL NETWORKS WITH ORTHOGONAL PROJECTION LAYERS

**Panagiotis D. Grontas**\*
ETH Zürich
pgrontas@ethz.ch

**Antonio Terpin**\*
ETH Zürich
aterpin@ethz.ch

**Efe C. Balta**
inspire AG & ETH Zürich
efe.balta@inspire.ch

**Raffaello D'Andrea**
ETH Zürich
rdandrea@ethz.ch

**John Lygeros**
ETH Zürich
jlygeros@ethz.ch

## ABSTRACT

We introduce an output layer for neural networks that ensures satisfaction of convex constraints. Our approach, Πnet, leverages operator splitting for rapid and reliable projections in the forward pass, and the implicit function theorem for backpropagation. We deploy Πnet as a *feasible-by-design* optimization proxy for parametric constrained optimization problems and obtain modest-accuracy solutions faster than traditional solvers when solving a single problem, and significantly faster for a batch of problems. We surpass state-of-the-art learning approaches by orders of magnitude in terms of training time, solution quality, and robustness to hyperparameter tuning, while maintaining similar inference times. Finally, we tackle multi-vehicle motion planning with non-convex trajectory preferences and provide Πnet as a GPU-ready package implemented in JAX.

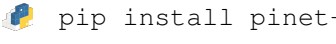 https://github.com/antonioterpin/pinet      pip install pinet-hcnn

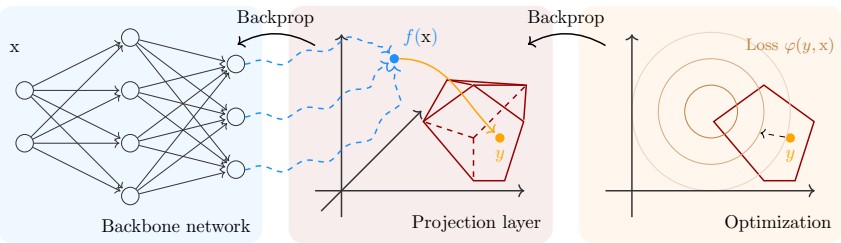

Figure 1: Illustration of the Πnet architecture. The infeasible output of the *backbone* network is projected onto the feasible set through an operator splitting scheme. To train the backbone network, we use the implicit function theorem to backpropagate the loss through the projection layer.

## 1 INTRODUCTION

In this work, we deploy neural networks (NNs) to generate *feasible-by-design* candidate solutions (see Figure 1) for the parametric constrained optimization problem:

$$\underset{y}{\text{minimize}} \quad \varphi(y, \mathrm{x}) \quad \text{subject to} \quad y \in \mathcal{C}(\mathrm{x}), \qquad\qquad \mathcal{P}(\mathrm{x})$$

where $y \in \mathbb{R}^d$ is the decision variable, $\mathrm{x} \in \mathbb{R}^p$ is the context (or parameter) of the problem instance, $\varphi : \mathbb{R}^d \times \mathbb{R}^p \to \mathbb{R}$ is the objective function, and $\mathcal{C}(\mathrm{x}) \subseteq \mathbb{R}^d$ is a non-empty, closed, convex set for all $\mathrm{x}$. We provide a pedagogical example to explain this formulation in Appendix A.

Constrained optimization has universal applicability, from safety-critical applications such as the optimal power flow in electrical grids (Nellikkath & Chatzivasileiadis, 2022), to logistics and scheduling (Bengio et al., 2021), and even biology, where enforcing priors on the solution can enhance its

---
\*Equal contribution.

interpretability (Balcerak et al., 2022; Terpin et al., 2024b). In many applications, optimization programs are solved repeatedly, given different contexts: in logistics, the demands and forecasts vary (Baptiste et al., 2001); in model predictive control (MPC) (Chen et al., 2018; Tabas & Zhang, 2022), the initial conditions; in motion planning, the position of obstacles (Marcucci et al., 2023); in trust-region policy optimization, the advantage function and the trust-region center (Terpin et al., 2022b). This task often becomes computationally challenging when, for example, $y$ is high-dimensional, $\varphi$ is non-convex, or new solutions are required at a high frequency. Rather than solving each problem instance from scratch, the mapping from contexts to solutions can be learned with NNs. Despite their successes, NNs typically lack inherent mechanisms to ensure satisfaction of explicit constraints, a limitation that motivates a large body of existing work:

**Soft-constrained NNs.** Soft penalty terms for constraint violations in the objective function are one approach to incorporate constraints in NNs (Márquez-Neila et al., 2017), and have been adopted to solve parametric constrained optimization problems (Tuor et al., 2021), and partial differential equations through physics-informed NNs (Erichson et al., 2019; Raissi et al., 2019). Despite their ability to handle general constraints, these approaches offer no guarantees at inference time. Beside requiring manual tuning of the penalty parameters, which is challenging yet critical for good performance, the use of soft constraints is discouraged for the following reasons. First, the structure of the constraints set can be exploited to design more efficient algorithms; see, e.g., the simplex algorithm (Dantzig, 2002). Second, treating constraints softly may significantly alter the problem solution regardless of tuning (Grontas et al., 2024). Third, certain constrained optimization problems (e.g., linear programs) may not admit a solution at all when constraints are treated softly.

**Hard-constrained NNs.** To circumvent the shortcomings of soft constraints, hard-constrained neural networks (HCNNs) aim to enforce constraints on the NN output by design. Frerix et al. (2020) address linear homogeneous inequality constraints by parameterizing the feasible set. Similarly, RAYEN (Tordesillas et al., 2023) enforces various convex constraints by scaling the line segment between infeasible points and a fixed point in the feasible set's interior. While these methods enjoy rapid inference, they require expensive offline preprocessing and are not directly applicable to constraints that depend on the NN input, i.e., they consider feasible sets $\mathcal{C}$ and not $\mathcal{C}(\mathrm{x})$ in $\mathcal{P}(\mathrm{x})$. Differently, Min et al. (2024) propose a closed-form expression to recover feasibility for polyhedral constraints and employs `cvxpylayers` (Agrawal et al., 2019) for more general convex sets. Chen et al. (2018); Cristian et al. (2023) orthogonally project the NN output or intermediate layers using Dykstra's algorithm (Boyle & Dykstra, 1986), but rely on loop unrolling for backpropagation, which can be prohibitive in terms of memory and computation. Departing from convex sets, `DC3` (Donti et al., 2021) introduces an equality completion and inequality correction procedure akin to soft-constrained approaches, but applied during inference. Lastrucci & Schweidtmann (2025) impose non-linear equality constraints by recursively linearizing them. Lagrangian and augmented Lagrangian approaches are considered by Fioretto et al. (2021); Park & Van Hentenryck (2023) for general non-convex constraints, with drawbacks similar to soft constraints, whereas Kratsios et al. (2021) consider a (probabilistic) sampling approach. Recently, LinSATNet (Wang et al., 2023) has been proposed to impose non-negative linear constraints, which is a restrictive constraint class that renders none of the problems of interest for this work amenable to LinSATNet. This limitation is partially relaxed by GLinSAT (Zeng et al., 2024), which however requires bounded constraints, an assumption not satisfied by, e.g., epigraph reformulations (Stellato et al., 2020, Appendix A.5-A.7).

**Implicit layers.** *Implicit layers* embed optimization problems such as quadratic programs (QPs) (Amos & Kolter, 2017; Butler & Kwon, 2023), conic programs (Agrawal et al., 2019), non-linear least-squares (Pineda et al., 2022), or fixed-point equations (Bai et al., 2019; Winston & Kolter, 2020) as NN layers, and apply the implicit function theorem (Dontchev & Rockafellar, 2009) to various measures of optimality. Our work is an instance of implicit layers; see Appendix C.5.

**ML for optimization.** Using constrained NN architectures to learn solution mappings is among many successful efforts to exploit ML techniques to accelerate (King et al., 2024; Sambharya et al., 2024) or replace optimization solvers (Bertsimas & Stellato, 2022; Zamzam & Baker, 2020). These approaches are referred to as *amortized optimization*, *learning to optimize* or *optimization learning*, and the surveys (Amos, 2023; Van Hentenryck, 2025) cover multiple aspects of the topic.

**Contributions.** We propose an NN architecture, Πnet, that generates feasible-by-design solutions for $\mathcal{P}(\mathrm{x})$. Given a context x, we deploy a *backbone* NN to produce a raw output $y_{\mathrm{raw}}$ that we orthogonally project onto $\mathcal{C}(\mathrm{x})$, $y = \mathrm{argmin}_{z \in \mathcal{C}(\mathrm{x})} \|z - y_{\mathrm{raw}}\|^2$. In particular:

- We use an operator splitting scheme to compute the projection in the forward pass, and back-propagate through it via the implicit function theorem. Our work is an instance of implicit layers (see Appendix C.5), but it specializes in projection problems whose structure can be significantly exploited, achieving rapid training and improved inference speed. Our simple idea yields a learning architecture that surpasses state-of-the-art learning approaches by orders of magnitude in terms of training time, solution quality, and robustness to hyperparameter tuning, while maintaining similar inference times.

- We implement a hyperparameter tuning and matrix equilibration strategy that boosts Πnet's performance and robustifies it against data scaling. As a result, Πnet solves challenging benchmarks on which existing methods struggle, and is substantially less sensitive to hyperparameter tuning. In addition, we deploy Πnet on a real-world application in multi-vehicle motion planning with non-convex trajectory costs.

- We provide an efficient and GPU-ready implementation of Πnet in JAX. We make our code available at https://github.com/antonioterpin/pinet.

**Notation.** The indicator function of a set $\mathcal{K}$ is $\mathcal{I}_{\mathcal{K}}(y) = 0$ if $y \in \mathcal{K}$, and $+\infty$ otherwise. The proximal operator of a proper, closed and convex function $f$ with parameter $\sigma > 0$ is $\mathrm{prox}_{\sigma f}(x) = \mathrm{argmin}_y \{f(y) + \frac{1}{2\sigma}\|y - x\|^2\}$. The projection onto a non-empty, closed, and convex set $\mathcal{C}$ is $\Pi_{\mathcal{C}}(x) = \mathrm{prox}_{\mathcal{I}_{\mathcal{C}}}(x)$. For a differentiable mapping $F : \mathbb{R}^n \to \mathbb{R}^m$, its Jacobian evaluated at $\hat{x} \in \mathbb{R}^n$ is denoted by $\frac{\partial F(x)}{\partial x}\big|_{x=\hat{x}} \in \mathbb{R}^{m \times n}$; we use d instead of $\partial$ when referring to total derivatives. Given a vector $v \in \mathbb{R}^m$, the vector-Jacobian product (VJP) of $F$ at $\hat{x}$ with $v$ is $v^\top \frac{\partial F(x)}{\partial x}\big|_{x=\hat{x}} \in \mathbb{R}^n$.

## 2 TRAINING HARD-CONSTRAINED NEURAL NETWORKS

We develop an NN layer that projects the output of any backbone NN onto $\mathcal{C}(\mathrm{x})$, and discuss how Πnet can be trained to generate solutions of $\mathcal{P}(\mathrm{x})$, for any given x.

### 2.1 PROJECTION LAYER

Given a context x, the backbone network produces the raw output $y_{\mathrm{raw}} = f(\mathrm{x}; \theta)$, where $\theta$ are the network weights. We enforce feasibility by projecting $y_{\mathrm{raw}}$ onto $\mathcal{C}(\mathrm{x})$:

$$y = \Pi_{\mathcal{C}(\mathrm{x})}(y_{\mathrm{raw}}) = \mathrm{argmin}_{z \in \mathcal{C}(\mathrm{x})} \|z - y_{\mathrm{raw}}\|^2. \tag{1}$$

We highlight two benefits of this approach:

✓ **Constraint satisfaction.** By design, the output $y$ of the projection layer always lies in $\mathcal{C}(x)$.

✓ **Decomposition of specifications.** The hard constraints prescribe the *required* behavior of the output, while the objective prescribes the *desired* behavior. Contrary to soft-constrained NNs, *no tradeoff between the two behaviors is introduced in our proposed framework*.

We consider constraints that can be expressed as $\mathcal{C} = \Pi_d(\mathcal{A} \cap \mathcal{K})$, where $\mathcal{A}, \mathcal{K} \subseteq \mathbb{R}^n$, $n \geq d$, are closed, convex sets that we design, and $\Pi_d$ is the projection onto the first $d$ coordinates. We omit the dependence on x for brevity, but stress that our method readily handles context-dependent constraints. Notice that we work in the possibly higher-dimensional $\mathbb{R}^n$, by introducing an auxiliary variable $y_{\mathrm{aux}} \in \mathbb{R}^{n-d}$. This provides us the flexibility to choose $\mathcal{A}$ and $\mathcal{K}$ such that their respective projections $\Pi_{\mathcal{A}}$ and $\Pi_{\mathcal{K}}$ admit a closed-form expression or are numerically-efficient. In particular, $\mathcal{A}$ will be a hyperplane defined by the coefficient matrix $A$ and the offset vector $b$, and $\mathcal{K}$ a Cartesian product of the form $\mathcal{K} = \mathcal{K}_1 \times \mathcal{K}_2 \subseteq \mathbb{R}^d \times \mathbb{R}^{n-d}$. This representation can describe many constraints of practical interest and we instantiate it with an example next (see Appendix E for more). Consider polytopic sets that are often employed in robotics (Chen et al., 2018), numerical solutions to partial differential equations (PDE) (Raissi et al., 2019), and non-convex relaxations for trajectory planning (Malyuta et al., 2022), among others. They are expressed as $\{y \in \mathbb{R}^d \mid Ey = q, l \leq Cy \leq u\}$, for

some $E, q, l, C, u$ of appropriate dimensions. We introduce the auxiliary variable $y_{\text{aux}} = Cy \in \mathbb{R}^{n_{\text{ineq}}}$ with dimension $n - d = n_{\text{ineq}}$. Then, we define $\mathcal{A}, \mathcal{K} \subseteq \mathbb{R}^n$ as the following hyperplane and box

$$\mathcal{A} = \left\{ \begin{bmatrix} y \\ y_{\text{aux}} \end{bmatrix} \; \middle| \; \underbrace{\begin{bmatrix} E & 0 \\ C & -I \end{bmatrix}}_{=A} \begin{bmatrix} y \\ y_{\text{aux}} \end{bmatrix} = \underbrace{\begin{bmatrix} q \\ 0 \end{bmatrix}}_{=b} \right\}, \quad \mathcal{K} = \left\{ \begin{bmatrix} y \\ y_{\text{aux}} \end{bmatrix} \; \middle| \; y \in \mathbb{R}^d, \; l \le y_{\text{aux}} \le u \right\}.$$

Importantly, $\mathcal{C} = \Pi_d(\mathcal{A} \cap \mathcal{K})$ and both $\Pi_{\mathcal{A}}$ and $\Pi_{\mathcal{K}}$ can be evaluated in closed form. More generally, the proposed decomposition, with $\Pi_{\mathcal{A}}$ and $\Pi_{\mathcal{K}}$ admitting closed-form expressions or being numerically efficient, is applicable to any constraint set of the form:

$$
\begin{aligned}
& A_{\text{eq}} y = b_{\text{eq}} && \text{(Equality)} \\
& \ell_{\text{box}} \le y \le u_{\text{box}} && \text{(Box)} \\
& \ell_{\text{ineq}} \le A_{\text{ineq}} y \le u_{\text{ineq}} && \text{(Inequality)} \\
& \|A_{\text{soc},i} y + a_{\text{soc},i}\|_2 \le f_{\text{soc},i}^\top y + b_{\text{soc},i}, \quad i = 1, \ldots, N_{\text{soc}} && \text{(SOC)} \\
& \|A_{\ell_1,i} y + a_{\ell_1,i}\|_1 \le f_{\ell_1,i}^\top y + b_{\ell_1,i}, \quad i = 1, \ldots, N_{\ell_1} && (\ell_1 - \text{ball}) \\
& \|A_{\ell_\infty,i} y + a_{\ell_\infty,i}\|_\infty \le f_{\ell_\infty,i}^\top y + b_{\ell_\infty,i}, \quad i = 1, \ldots, N_{\ell_\infty} && (\ell_\infty - \text{ball}) \\
& y_{\mathcal{S}_j} \ge 0, \quad \|y_{\mathcal{S}_j}\|_1 = 1, \quad \text{for index sets } \mathcal{S}_j. && \text{(Simplex)}
\end{aligned}
\tag{2}
$$

**Remark.** *We stress that the decomposition $\mathcal{C} = \Pi_d(\mathcal{A} \cap \mathcal{K})$ is not an assumption. One can always decompose a convex set in this way, e.g., by considering the trivial decomposition $\mathcal{A} = \mathcal{C}$ and $\mathcal{K} = \mathbb{R}^d$. Instead, determining $\mathcal{A}$ and $\mathcal{K}$ is a design choice which we leverage to make the projections $\Pi_{\mathcal{A}}$ and $\Pi_{\mathcal{K}}$ computationally efficient. We note two important points regarding this design choice:*

- *The only assumption is that $\Pi_{\mathcal{A}}$ and $\Pi_{\mathcal{K}}$ and their VJP are computable. Being computationally efficient is an added benefit of our decomposition, but is not necessary. In this work, our focus is on the sets listed as those are the ones that are most interesting in practice.*

- *We show in Appendix E that many practically-relevant constraints $\mathcal{C}$, such as the ones listed in (2), as well as their intersections and Cartesian products all admit efficient decompositions. In fact, this list is not exhaustive; see, e.g., the work of Condat (2016) and Boyd & Vandenberghe (2004).*

### 2.1.1 FORWARD PASS

To compute the projection $y = \Pi_{\mathcal{C}(\text{x})}(y_{\text{raw}})$ we employ the Douglas-Rachford algorithm (Bauschke & Combettes, 2017, Sec. 28.3), which solves optimization problems of the form $\min_z g(z) + h(z)$, where $g$ and $h$ are proper, closed, convex functions. To rewrite (1) in this composite form, we use the auxiliary variable $y_{\text{aux}} \in \mathbb{R}^{n-d}$ and the indicator function of $\mathcal{A}$ and $\mathcal{K}$ to obtain:

$$(\Pi_{\mathcal{C}}(y_{\text{raw}}), y_{\text{aux}}^\star) = \operatorname*{argmin}_{y, y_{\text{aux}}} \left\{ \|y - y_{\text{raw}}\|^2 + \mathcal{I}_{\mathcal{A}}\left( \begin{bmatrix} y \\ y_{\text{aux}} \end{bmatrix} \right) + \mathcal{I}_{\mathcal{K}}\left( \begin{bmatrix} y \\ y_{\text{aux}} \end{bmatrix} \right) \right\}. \tag{3}$$

Then, we split the objective function as follows

$$g\left( \begin{bmatrix} y \\ y_{\text{aux}} \end{bmatrix} \right) = \mathcal{I}_{\mathcal{A}}\left( \begin{bmatrix} y \\ y_{\text{aux}} \end{bmatrix} \right) \quad \text{and} \quad h\left( \begin{bmatrix} y \\ y_{\text{aux}} \end{bmatrix} \right) = \|y - y_{\text{raw}}\|^2 + \mathcal{I}_{\mathcal{K}}\left( \begin{bmatrix} y \\ y_{\text{aux}} \end{bmatrix} \right).$$

By applying the Douglas-Rachford algorithm we obtain the fixed-point iteration:

$$
\begin{aligned}
z_{k+1} &= \operatorname{prox}_{\sigma g}(s_k) && \text{(4a)} \\
t_{k+1} &= \operatorname{prox}_{\sigma h}(2z_{k+1} - s_k) && \text{(4b)} \\
s_{k+1} &= s_k + \omega(t_{k+1} - z_{k+1}) && \text{(4c)}
\end{aligned}
$$

where $\sigma > 0$ is a scaling and $\omega \in (0, 2)$ a relaxation parameter. The proximal operators in (4a) and (4b) can be evaluated explicitly, see Appendix D.1, allowing us to implement (4) as in Algorithm 1. Note that we write $z_k = \begin{bmatrix} z_{k,1} & z_{k,2} \end{bmatrix}^\top$ where $z_{k,1} \in \mathbb{R}^d$ and $z_{k,2} \in \mathbb{R}^{n-d}$ correspond to $y$ and $y_{\text{aux}}$.

---

**ALGORITHM 1. Operator splitting for projection**

**Inputs:** $\text{x}, y_{\text{raw}}, K \in \mathbb{N}, \sigma, \omega$.
**Initialization:** $s_0 \in \mathbb{R}^n$
**For** $k = 0$ **to** $K - 1$**:**

$\quad z_{k+1} \leftarrow \Pi_{\mathcal{A}(\text{x})}(s_k)$

$\quad t_{k+1} \leftarrow \begin{bmatrix} \Pi_{\mathcal{K}_1(\text{x})}\left( \frac{2z_{k+1,1} - s_{k,1} + 2\sigma y_{\text{raw}}}{1 + 2\sigma} \right) \\ \Pi_{\mathcal{K}_2(\text{x})}(2z_{k+1,2} - s_{k,2}) \end{bmatrix}$

$\quad s_{k+1} \leftarrow s_k + \omega(t_{k+1} - z_{k+1})$

**Output:** $z_{K,1}, s_K$

---

Under mild conditions, namely strict feasibility of (3), we show in Appendix D.2 that the iterates $z_k$ and $t_k$ converge to a solution of (3). We denote the limits $z_\infty(y_{\text{raw}}) = \lim_{k \to \infty} z_k$ and $s_\infty(y_{\text{raw}}) = \lim_{k \to \infty} s_k$, and highlight their dependence on the point-to-be-projected $y_{\text{raw}}$. In particular, we have $z_{\infty,1}(y_{\text{raw}}) = \Pi_{\mathcal{C}}(y_{\text{raw}})$. In practice, we run a finite number of iterations $K \in \mathbb{N}$ of (4), which we set to $K = \texttt{n\_iter\_fwd}$ during training and $K = \texttt{n\_iter\_test}$ during testing, and take $y = z_{K,1}$ as the output of the projection layer. We detail our hyperparameters in Section 2.4. We note that, although $z_K$ will not necessarily lie on $\mathcal{A} \cap \mathcal{K}$, because $K \in \mathbb{N}$ is finite, Algorithm 1 guarantees that $z_K \in \mathcal{A}$. By our choice of $\mathcal{A}$, this implies that $z_{K,1}$ satisfies any equality constraints in the problem. The feasibility-by-design comes from the convergence rates, which we derive and empirically demonstrate in Appendix D.3: for a sufficiently high number of iterations, the output of our projection layer is arbitrarily close to the true projection.

### 2.1.2 BACKWARD PASS

To train the backbone network using backpropagation, we need to efficiently differentiate the loss $\mathcal{L}$ (which in general depends on the projected output of the network and on the input; see Section 2.2) with respect to the backbone network parameters $\theta$. The chain rule gives us:

$$\frac{\mathrm{d}\mathcal{L}(\Pi_{\mathcal{C}}(f(\mathrm{x};\theta)), \mathrm{x})}{\mathrm{d}\theta} = \left.\frac{\partial\mathcal{L}(y, \mathrm{x})}{\partial y}\right|_{y = \Pi_{\mathcal{C}}(f(\mathrm{x};\theta))} \left.\frac{\partial\Pi_{\mathcal{C}}(y_{\text{raw}})}{\partial y_{\text{raw}}}\right|_{y_{\text{raw}} = f(\mathrm{x};\theta)} \left.\frac{\partial f(\mathrm{x};\vartheta)}{\partial \vartheta}\right|_{\vartheta = \theta}. \quad (5)$$

Since the first and last terms are standard and typically computed with automatic differentiation, we only need to provide an efficient computational routine for the VJP

$$v \mapsto v^\top \left(\left.\frac{\partial\Pi_{\mathcal{C}}(y_{\text{raw}})}{\partial y_{\text{raw}}}\right|_{y_{\text{raw}} = f(\mathrm{x};\theta)}\right). \quad (6)$$

Rather than differentiating through all the iterations of Algorithm 1 by loop unrolling, we exploit the implicit function theorem (Dontchev & Rockafellar, 2009) to efficiently evaluate (6). To do so, we first recall that $\Pi_{\mathcal{C}}(y_{\text{raw}}) = \begin{bmatrix} I_d & \mathbf{0}_{d \times (n-d)} \end{bmatrix} \Pi_{\mathcal{A}}(s_\infty(y_{\text{raw}}))$ which implies that:

$$v^\top \left(\left.\frac{\partial\Pi_{\mathcal{C}}(y_{\text{raw}})}{\partial y_{\text{raw}}}\right|_{y_{\text{raw}} = f(\mathrm{x};\theta)}\right) = \left(\begin{bmatrix} v^\top & \mathbf{0} \end{bmatrix} \left.\frac{\partial\Pi_{\mathcal{A}}(s)}{\partial s}\right|_{s = s_\infty(y_{\text{raw}})}\right) \left.\frac{\partial s_\infty(y_{\text{raw}})}{\partial y_{\text{raw}}}\right|_{y_{\text{raw}} = f(\mathrm{x};\theta)}, \quad (7)$$

and note that the first VJP is straightforward to compute since $\Pi_{\mathcal{A}}$ is an affine mapping. Therefore, to evaluate (7) we need to backpropagate through $s_\infty(y_{\text{raw}})$. Since $s_\infty(y_{\text{raw}})$ is the fixed point of iteration (4), it satisfies the equation $s_\infty(y_{\text{raw}}) = \Phi(s_\infty(y_{\text{raw}}), y_{\text{raw}})$, where $\Phi$ represents one iteration of (4a)-(4c), namely, $s_{k+1} = \Phi(s_k, y_{\text{raw}})$. The implicit function theorem applied to $s_\infty(y_{\text{raw}}) = \Phi(s_\infty(y_{\text{raw}}), y_{\text{raw}})$ yields the VJP, see Appendix D.4,

$$v \mapsto \xi(y_{\text{raw}}, v)^\top \left.\frac{\partial\Phi(s_\infty(y_{\text{raw}}), y_{\text{raw}})}{\partial y_{\text{raw}}}\right|_{y_{\text{raw}} = f(\mathrm{x};\theta)}, \quad (8)$$

where $\xi(y_{\text{raw}}, v)$ is a solution of the linear system

$$\left(I - \left.\frac{\partial\Phi(s, y_{\text{raw}})}{\partial s}\right|_{s = s_\infty(y_{\text{raw}})}\right)^\top \xi(y_{\text{raw}}, v) = v. \quad (9)$$

The matrix in (9) may not be invertible (Agrawal et al., 2019); even if it is, constructing it and computing its inverse may be prohibitively expensive in high dimensions. This difficulty can be circumvented by computing a heuristic quantity. In this vein, we deploy the JAX (Bradbury et al., 2018) implementation of the bi-conjugate gradient stable iteration (`bicgstab`, van der Vorst (1992)), an indirect linear system solver that requires only matrix-vector products. Therefore, we implement (8) and (9) using VJPs involving $\partial\Phi(s_\infty(y_{\text{raw}}), y_{\text{raw}})/\partial y_{\text{raw}}$ and $\partial\Phi(s, y_{\text{raw}})/\partial s$, respectively. We efficiently do this using JAX VJP routines and note that each step of the solver for (9) has essentially the same computational cost as one step of (4). The maximum number of `bicgstab` steps for (9), `n_iter_bwd`, is a hyperparameter. In both (8) and (9), we use the output of the forward pass, i.e., $s_K$ in place of the intractable $s_\infty$. We discuss almost-everywhere differentiability of our projection layer and the applicability of the implicit function theorem in Appendix F.

## 2.2 Loss

The training loss $\mathcal{L}\big(\Pi_{\mathcal{C}(\mathrm{x})}(f(\mathrm{x};\theta)),\mathrm{x}\big)$, which is a function of the context and the constrained NN output, can be crafted according to the specific requirements of the problem. In our experiments, we directly minimize the objective of $\mathcal{P}(\mathrm{x})$ using the network's output by setting $\mathcal{L}(y,\mathrm{x}) = \varphi(y,\mathrm{x})$. In our framework we can, loosely speaking, interpret training as performing projected gradient descent on the raw NN output space, akin to the work of Cristian et al. (2023).

## 2.3 Network architecture

$\Pi$net is flexible: the projection layer can be appended to *any* NN; see Figure 1. We summarize the forward and backward pass of $\Pi$net in Alg. 2, while its training and testing is outlined in Alg. 3.

---

**ALGORITHM 2. $\Pi$net**

`Forward:`
> **Inputs:** $\mathrm{x}, \theta, K, \sigma, \omega$
> $y_{\mathrm{raw}} \leftarrow f(\mathrm{x};\theta)$
> $y, s \leftarrow$ Call Algorithm 1$(\mathrm{x}, y_{\mathrm{raw}}, K, \sigma, \omega)$
> **Output:** $y, s$

`Backward:`
> **Inputs:** $s_\infty, v \in \mathbb{R}^d$
> $v_1 \leftarrow \begin{bmatrix} v^\top & \mathbf{0}\end{bmatrix}(\partial\Pi_{\mathcal{A}}(s)/\partial s)\big|_{s=s_\infty}$
> $\xi(y_{\mathrm{raw}}, v_1) \leftarrow$ Solve (9) with $v_1$ as RHS
> $v_2 \leftarrow \xi(y_{\mathrm{raw}}, v_1)^\top(\partial\Phi(s, y_{\mathrm{raw}})/\partial y_{\mathrm{raw}})\big|_{s=s_\infty}$
> $v_3 \leftarrow v_2^\top(\partial f(\mathrm{x};\vartheta)/\partial\vartheta)\big|_{\vartheta=\theta}$
> **Output:** $v_3$

---

**ALGORITHM 3. Training/Testing of $\Pi$net**

**Inputs**: Chosen/Tuned hyperparameters
    (see Section 2.4)
`Training:`
> $\theta_1 \leftarrow$ Initialize weights
> **For $\ell = 1$ to n_epochs:**
>> $\mathrm{x} \leftarrow$ Sample from training set
>> $y, s \leftarrow$ `Forward`$(\mathrm{x}, \theta, \texttt{n\_iter\_fwd}, \sigma, \omega)$
>> $\mathcal{L}(y,\mathrm{x}), \partial\mathcal{L}(y,\mathrm{x})/\partial y \leftarrow$ Compute loss
>> $\partial\mathcal{L}/\partial\theta \leftarrow$ `Backward`$(s, \partial\mathcal{L}(y,\mathrm{x})/\partial y)$
>> $\theta_{\ell+1} \leftarrow$ Optimizer update
>
> **Output** : $\theta_{\texttt{n\_epochs}}$

`Testing:`
> $\mathrm{x} \leftarrow$ Sample from test set
> $y, s \leftarrow$ `Forward`$(\mathrm{x}, \theta, \texttt{n\_iter\_test}, \sigma, \omega)$
> **Output** : $y$

---

## 2.4 The sharp bits (short version)

To push the performance of $\Pi$net, we adopt two important numerical techniques. First, we improve the conditioning of the matrix $A(\mathrm{x})$, that defines the set $\mathcal{A}(\mathrm{x})$, by implementing the Ruiz equilibration algorithm (Wathen, 2015); see Appendix C.1. Second, we exploit the fact that, compared to existing methods, $\Pi$net relies only on a few hyperparameters. Specifically, n_iter_fwd (number of iterations for the forward pass during training), n_iter_test (number of iterations for the forward pass during inference), omega, sigma (standard Douglas-Rachford parameters), and n_iter_bwd (number of iterations in the bicgstab procedure). We describe an auto-tuning procedure that recommends hyperparameters by evaluating the projection on a subset of the validation set in Appendix C.2, and assess its effectiveness in Appendix C.3. Finally, we highlight our choice of enforcing constraints during training, as opposed to training an unconstrained network and introducing the projection layer only afterwards. We do so because the latter approach may result in training instabilities or suboptimal performance; see Appendix C.4.

## 3 Numerical experiments

The code is made available at https://github.com/antonioterpin/pinet. The empirical data was collected on an Ubuntu 22.04 machine equipped with an AMD Ryzen Threadripper PRO 5995WX processor and an Nvidia RTX 4090 GPU. For experiments with second-order cone constraints, see Appendix B.5.

## 3.1 Benchmarks and comparisons with state-of-the-art

We consider a set of standard convex and non-convex problems classically used to compare HCNNs.

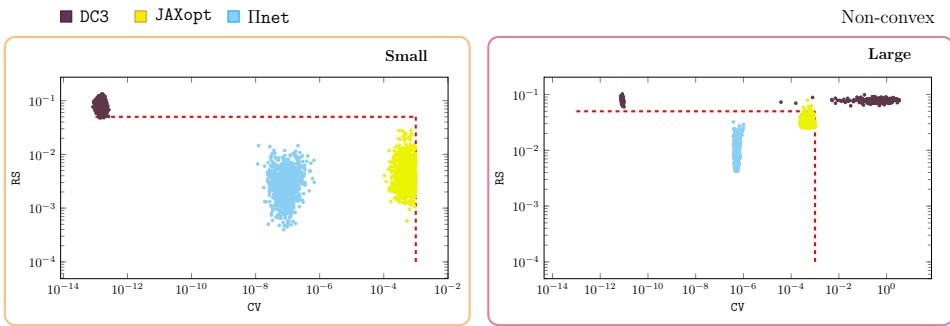

Figure 2: Scatter plots of RS and CV on the small and large non-convex problems on the test set. The red dashed lines show the thresholds to consider a candidate solution optimal.

**Experimental setup.** We consider the benchmark problems introduced by Donti et al. (2021):

$$\underset{y \in \mathbb{R}^d}{\text{minimize}} \quad J(y) \quad \text{subject to} \quad Ay = \mathrm{x}, \quad Cy \leq u,$$

where $J(\cdot)$ is defined as $J(y) = y^\top Qy + q^\top y$ for the convex problem setup, and as $J(y) = y^\top Qy + q^\top \sin(y)$ for the non-convex, and $Q \in \mathbb{R}^{d \times d} \succ 0$, $q \in \mathbb{R}^d$, $A \in \mathbb{R}^{n_{\mathrm{eq}} \times d}$, $\mathrm{x} \in \mathbb{R}^{n_{\mathrm{eq}}}$, $C \in \mathbb{R}^{n_{\mathrm{ineq}} \times d}$, $u \in \mathbb{R}^{n_{\mathrm{ineq}}}$. In particular, in the work of Donti et al. (2021), $Q$ is diagonal with positive entries, $A, C, u$ are fixed matrices/vector and the contexts x are generated so that all problem instances are guaranteed to be feasible. Donti et al. (2021) consider problems with $d = 100$. Here, we include larger problem dimensions, $(d, n_{\mathrm{eq}}, n_{\mathrm{ineq}}) \in \{(100, 50, 50), (1000, 500, 500)\}$, which we generated with the same scheme. We refer to these datasets as small and large. For each dataset, we generated 10000 contexts split as $7952/1024/1024$ among training/validation/test sets.

**Baselines.** We compare Πnet to DC3 (Donti et al., 2021) and a traditional Solver. For the convex objective the Solver is the QP solver OSQP (Stellato et al., 2020), while for the non-convex objective is IPOPT (Wächter & Biegler, 2006). Further, we compare to an implicit layer approach, where instead of computing the projection (1) with Algorithm 1, we use the JAXopt (Blondel et al., 2022) GPU-friendly implementation of OSQP that employs implicit differentiation. Both Πnet, DC3, and the JAXopt approach use a self-supervised loss (i.e., $\mathcal{L} = J$) and as backbone a multi-layer perceptron (MLP) with 2 hidden layers of 200 neurons each and ReLU activations. Additionally, DC3 includes batch normalization and drop out, as well as soft penalty terms in the loss. We use the default parameters of DC3 unless otherwise stated. In particular, the DC3 algorithm with default parameters diverged during training on the large datasets, an effect observed by Tordesillas et al. (2023). To rectify this, we tuned the learning rate of DC3's correction process for the large dataset, and found that $10^{-8}$ enables the network to learn. In Appendix B.2, we investigate if DC3's performance can be improved by adapting hyperparameters. For Πnet we use only 50 training epochs, while for DC3 we use the default 1000. For JAXopt we use a tolerance of $10^{-3}$ and 12 epochs, in the interest of training time. On both convex and non-convex benchmarks, we use JAXopt as a replacement for our custom projection layer after the backbone NN. The training times reported are, thus, the ones of the backbone network. We omit comparisons with cvxpylayers (Agrawal et al., 2019) since JAXopt is a more recent and stronger baseline: it implements similar functionalities, but it is executable on the GPU; see also Appendix B.

**Metrics.** We compare methods in terms of the following metrics on the test set:

- Relative suboptimality (RS): The suboptimality of a candidate solution $\hat{y}$ compared to the optimal objective $J(y^\star)$, computed by the Solver. Since methods may violate constraints and obtain a better solution we clip this value, $\text{RS} := \max\left(0, (J(\hat{y}) - J(y^\star))/J(y^\star)\right)$.

- Constraint violation (CV): We define $\text{CV} = \max(\|A\hat{y} - \mathrm{x}\|_\infty, \|\max(C\hat{y} - u, 0)\|_\infty)$.

- Learning curves: Progress on RS and CV over wall-clock time on the validation set.

- Single inference time: The time required to solve one instance at test time.

- Batch inference time: The time required to solve a batch of 1024 instances at test time.

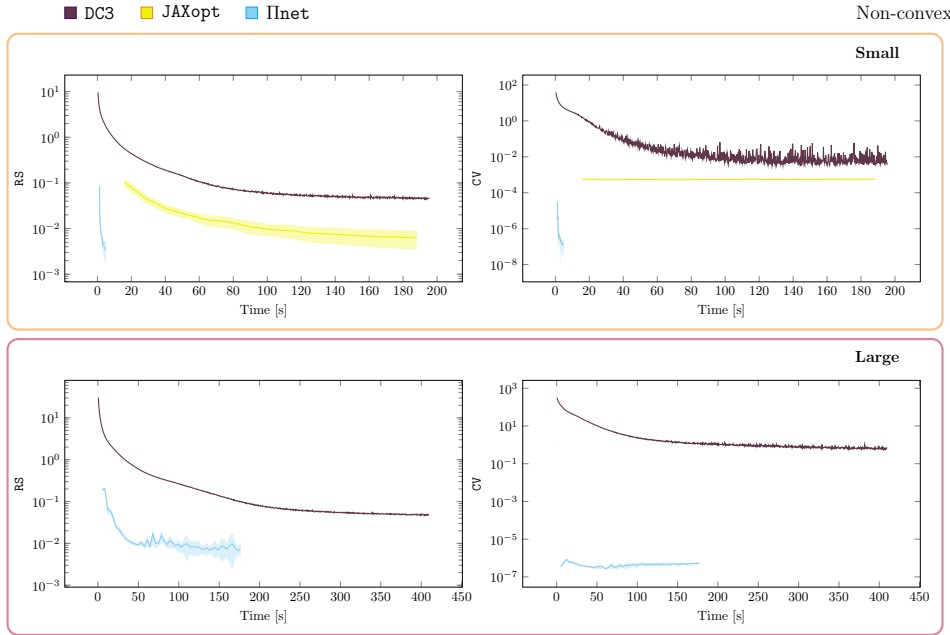

Figure 3: Comparison of the learning curves in terms of average `RS` and `CV` on the validation set, on the small and large non-convex problems. The solid lines denote the mean and the shaded area the standard deviation across 5 seeds. The learning curves for `JAXopt` on the large dataset are reported only in Appendix B because of the orders of magnitude longer training times.

Next, we report and discuss the results on the non-convex datasets. In the interest of space, the results on the convex problems are given in Appendix B.2.

**Results.** The `RS` and `CV` for each problem instance in the test set are reported in Figure 2. We consider a candidate solution to be optimal if the condition $CV \leq 10^{-3}$ and $RS \leq 5\%$ is satisfied. These prerequisites for low accuracy solutions are similar to, though somewhat looser from, those employed by numerical solvers (O'donoghue et al., 2016; Stellato et al., 2020). In fact, $\Pi$net clears these thresholds by a margin. In practice, any solver achieving a `CV` below $10^{-5}$ is considered high-accuracy (Stellato et al., 2020) and there is little benefit to go below that. Instead, when methods have sufficiently low `CV`, having a low `RS` is better.

Compared to `DC3`, we correctly solve the vast majority of test problems. Importantly, the very low and consistent constraint violation across all problem instances significantly facilitates the tuning of the number of iterations. By contrast, on the large problems, `DC3` exhibits `CV` and `RS` that are unacceptably large for any meaningful application. We conjecture that $\Pi$net significantly outperforms `DC3` in `RS` due to the absence of soft penalties in our training loss and the orthogonality of the projection. The `JAXopt` approach performs better than `DC3` but significantly worse than $\Pi$net.

The learning curves are shown in Figure 3. $\Pi$net achieves better performance at a fraction of the training time. Crucially, our scheme attains satisfactory `CV` *throughout* training, implying that $\Pi$net can reliably compute feasible solutions even with a tiny training budget. Note that our training curves include the setup time for $\Pi$net (the matrix equilibration, the calculation of the pseudo-inverse for the projection onto the affine subspace, and just-in-time compilation). We omit the `JAXopt` results on the large dataset as it requires roughly 14 hours to complete training. The substantial difference in training times between $\Pi$net and `JAXopt` underscores the importance of the specialized splitting we employed exploiting the structure of the projection problem, and our specialized implementation.

We report inference times in Table 2 in Appendix B. All approaches significantly outperform the `Solver`. Although `DC3` is slightly faster than $\Pi$net, it generates solutions of much lower quality (in `RS` and `CV`), hence demonstrating that the advantages of $\Pi$net come with only a minor runtime trade-off compared to `DC3`.

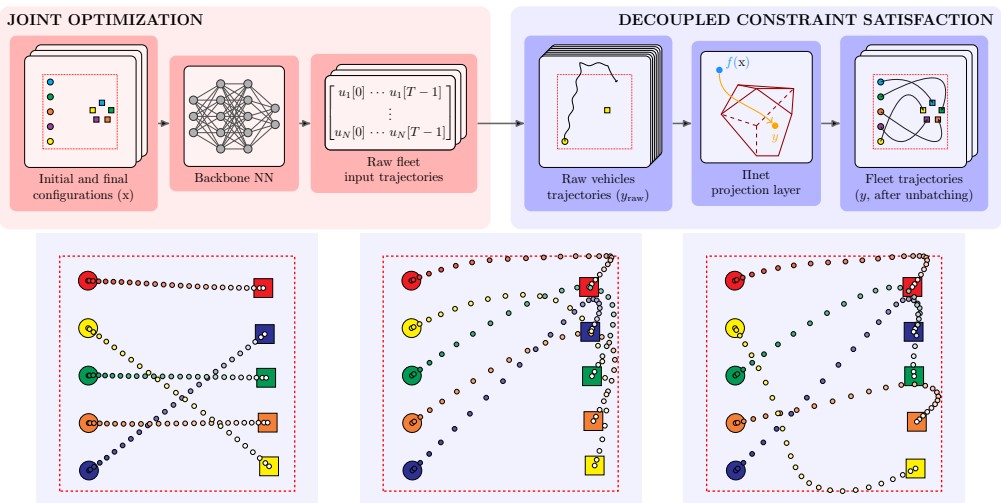

Figure 4: (Top) The Πnet approach to constrained multi-vehicle motion planning with arbitrary differentiable objective functions $\varphi$. (Bottom) From left to right, we show examples of the synthesized trajectories for 3 different objectives: $\varphi_{\text{left}} = \texttt{effort}, \varphi_{\text{mid}} = \varphi_{\text{left}} + \texttt{preference}, \varphi_{\text{right}} = \varphi_{\text{mid}} + \texttt{coverage}$. We refer the reader to Appendix B.3 for formal definitions and additional plots.

To summarize our findings, we showed that existing HCNN methods suffer either from long training/inference times or low quality solutions. Πnet address both shortcomings, while retaining modularity and simplicity.

## 3.2 ΠNET APPLIED: MULTI-VEHICLE MOTION PLANNING

We present an approach to synthesize transition trajectories between multi-vehicle configurations that optimize some non-linear, *fleet-level* objective subject to dynamics, state and input constraints. We feed an NN with the initial and terminal fleet configurations (the context x), obtain the raw input trajectories and use the vehicle dynamics to infer the full state-input trajectories that serves as the raw output $y_{\text{raw}}$, which are then projected for ensured constraint satisfaction; see Figure 4.

**Experimental setup.** We follow the formulation in the work of Augugliaro et al. (2012). Specifically, we denote with $p_i[t] \in \mathbb{R}^m$ the generalized coordinates of vehicle $i$ at the discrete times $t \in \{1, \ldots, T\}$, and with $v_i[t]$ and $a_i[t]$ its generalized velocity and acceleration. Its simple discretized dynamics read $v_i[t+1] = v_i[t] + ha_i[t], p_i[t+1] = p_i[t] + hv_i[t] + \frac{h^2}{2}a_i[t]$. We formulate the motion planning task for the fleet as a parametric program in the form of $\mathcal{P}(\text{x})$, where $\mathcal{C}(\text{x})$ includes box constraints on positions (workspace constraints), velocities, and accelerations (physical limits), affine inequality constraints for jerk limits, and equality constraints for the dynamics and initial/final configuration *for each vehicle*. The objective function $\varphi$ encapsulates a *fleet-level* objective; here we consider a weighted sum of workspace coverage, input effort, and trajectory preference given by a potential function; see Appendix B.3 for a rigorous definition.

**Qualitative results.** We display some of the resulting trajectories for different weights in the objective in Figure 4 and report more visualizations and analysis for larger fleets and longer horizons in Appendix B.3. Crucially, both convex and non-convex objectives are handled effectively by Πnet, resulting in trajectories that adhere to the goals prescribed by the different objective functions.

**Practical Relevance.** Multi-vehicle motion planning has received substantial attention for its practical applications (Terpin et al., 2022a; 2024a), and we highlight three benefits of our approach:

✓ **Constraint satisfaction.** We ensure dynamics, state, and input constraint satisfaction, similar to optimization-based trajectory generation methods (Augugliaro et al., 2012).

✓ **Parallelizability.** Our approach is parallelizable in two ways. First, it enables multiple problem instances (different initial and final configurations) to be solved in batches. Second, since the

constraints we consider are decoupled between the vehicles, we can *jointly* predict the raw input trajectories (to enable the network to minimize the joint objective), while solving the projections for each vehicle separately; see Figure 4.

✓ **Arbitrary objective optimization.** Our framework can handle any almost everywhere differentiable objective, encoded in Πnet's loss $\mathcal{L} = \varphi$. Importantly, we see this example as a proof of concept towards constrained human-preference optimization (e.g., using the approach of Christiano et al. (2017)). Deploying a traditional solver for this problem is very challenging because the objective functions considered are not available in closed form and are highly non-linear.

We implement this application in a separate codebase `https://github.com/antonioterpin/glitch`, demonstrating also the little overhead required to integrate Πnet into specific applications. We also explore trajectory planning on a longer horizon (up to 750 steps, amounting to about 9000 optimization variables and constraints), as well as high-dimensional contexts (mono-channel, $1024 \times 1024$ images) in Appendix B.4. Our current formulation focuses only on convex, decoupled-among-vehicles constraints. Future works could address collision avoidance constraints through sequential convexification techniques (Augugliaro et al., 2012; Malyuta et al., 2022).

## 4 CONCLUSIONS

**Contributions.** We introduced an output layer that enforces convex constraints satisfaction on the output of an any backbone NN via an operator splitting scheme. The backpropagation is achieved via the implicit function theorem, enabling efficient training. Our work focused on the gritty details of optimizing Πnet, introducing also simple yet effective techniques such as hyperparameter tuning and matrix equilibration procedures. We showed through extensive benchmarks that Πnet succeeds where existing learning methods fail. We provide a GPU-ready implementation in JAX, and showcase how our layer can be embedded in an example application, multi-vehicle motion planning.

**Limitations.** One limitation of our work is the requirement of convex constraint sets. Despite the numerous applications involving only convex constraints (Boyd & Vandenberghe, 2004), and the numerous applications that can be losslessly convexified (Malyuta et al., 2022), we acknowledge that future work should investigate how to relax this structural assumption. One potential approach could involve sequential convexification of non-convex constraints, similar to the algorithm in (Lastrucci & Schweidtmann, 2025) that addresses non-linear equality constraints, or by employing homeomorphisms (Liang et al., 2024).

**Outlook.** We believe that Πnet holds the potential to substantially impact a wide range of machine learning domains where constraint satisfaction is crucial. Relevant examples include neural PDE solvers (Raissi et al., 2019), scheduling (Baptiste et al., 2001), and robotics (Malyuta et al., 2022), among others. We demonstrated the potential of Πnet in some of these applications in Section 3, and we believe that applying our method to new applications represents an exciting avenue for future work. We expect the integration of hard constraints into large-scale models to result in more robust performance and more trustworthy machine learning systems.

## ACKNOWLEDGMENTS AND DISCLOSURE OF FUNDING

We would like to thank the reviewers for their constructive comments. This work was supported as a part of NCCR Automation, a National Centre of Competence in Research, funded by the Swiss National Science Foundation (grant number 51NF40_225155).

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

## CONTENTS

LIST OF FIGURES

LIST OF TABLES

# A  TOY VISUAL EXAMPLE

In this section, we instantiate our method on a very simple and interpretable toy problem: learning to predict the outputs of a model predictive control (MPC) policy (Chen et al., 2018). In high-dimensional scenarios and for long horizons, running MPC in real-time often becomes a computational bottleneck, so learning to predict its outputs–even with minor suboptimality, as long as constraints are still satisfied–can bring substantial benefits (Chen et al., 2018).

**Experimental setting.**  We consider a two-dimensional single integrator system, and define an MPC control law given by the solution of

$$\underset{u_k \in [-1,1]^2}{\text{minimize}} \sum_{k=0}^{N-1} \|x_k - \hat{x}\|^2 + \|u_k\|^2 \quad \text{subject to} \quad x_0 = \text{x}, \ x_{k+1} = x_k + u_k, \ x_k \in [-10, 10]^2. \quad (10)$$

Here, $x_k$ denotes the system state, $u_k$ the control input, and $\hat{x} = [3, -12]$ is a target state. We sample the context (in this case, the initial condition for the MPC), $\text{x} = x_0$, uniformly in $[-10, 10]^2$. We deploy $\Pi$net to learn the solution of (10) as a function of x. We use a self-supervised loss and the backbone NN is an MLP with two hidden layers of 200 neurons and `ReLU` activations.

**Results.**  We superimpose $\Pi$net's prediction and the solution of (10) given by a `Solver`. Further, to illustrate the importance of *hard* constraints, we also plot the prediction of the same MLP trained with soft constraints, i.e., adding to the loss the term $\lambda(\texttt{EQCV} + \texttt{INEQCV})$, where `EQCV` and `INEQCV` is the maximum violation of the equality and inequality constraints, respectively, for two values of $\lambda$.

We represent $\hat{x}$ with a star, x with a square, and state constraints with a dashed rectangle. We tested the trained network in 1000 instances generated by sampling x uniformly in $[-10, 10]^2$. $\Pi$net achieves an average relative suboptimality of approximately 0.02%, and constraints satisfied in 100% of problem instances to within a tolerance of $10^{-3}$. It is also apparent that different values of $\lambda$ induce different behaviors: Larger values enforce the constraints at the expense of optimality, smaller values do not enforce constraints.

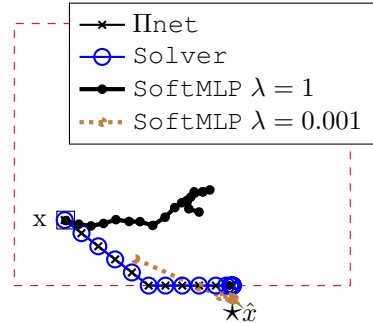

| Method | RS | CV | Single inference time [s] | Batch inference time [s] |
|---|---|---|---|---|
| cvxpylayers | 0.0036 | 0.00005 | 0.0120 | 2.5917 |
| Πnet (CPU) | 0.0035 | 0.00000 | 0.0052 | 0.0957 |
| Πnet (GPU) | 0.0035 | 0.00000 | 0.0065 | 0.0135 |

Table 1: Comparison with `cvxpylayers` on the small, non-convex benchmark. The `RS` and `CV` values reported are the averages over the test set.

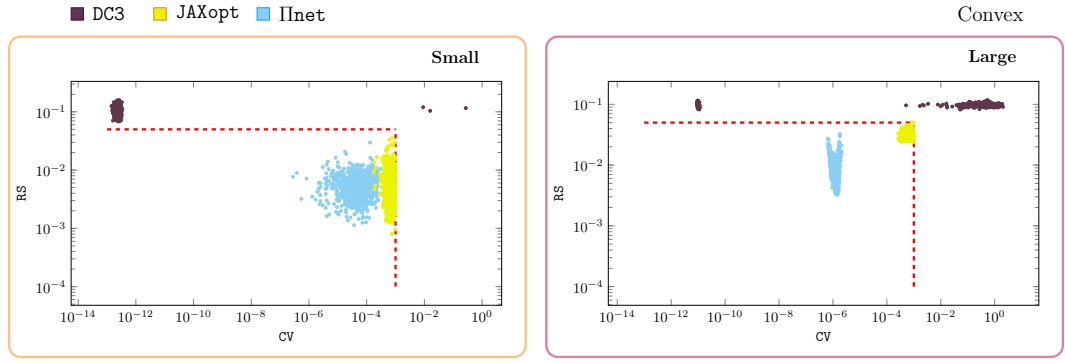

Figure 5: Scatter plots of `RS` and `CV` on the small and large convex problems on the test set. The red dashed lines show the thresholds to consider a candidate solution optimal.

## B  ADDITIONAL RESULTS

In this section, we collect the additional results for Section 3 and the ablation studies on some `DC3` hyperparameters.

### B.1  A COMPARISON WITH `CVXPYLAYERS`.

We compare Πnet with `cvxpylayers` on the small non-convex benchmark, and report the results in Table 1. We use `cvxpylayers` as an alternative to our custom projection layer, in an analogous manner to the comparison with `JAXopt` in Section 3.1. Since `cvxpylayers` runs exclusively on the CPU, we report the results of Πnet on both CPU and GPU. In all cases, we provide a training budget of 25 epochs. For `cvxpylayers` this corresponds to roughly 10 minutes of wall-clock time, whereas for Πnet this corresponds to 4 and 28 seconds on the GPU and CPU, respectively. We observe that Πnet attains similar performance in terms of `RS` and `CV`, but significantly outperforms `cvxpylayers` in terms of inference and training time. We omit further results with `cvxpylayers`, since `JAXopt` is a more recent and stronger baseline: it implements similar functionalities, but it is executable on the GPU (see Figures 2 and 3, and Table 2).

### B.2  ADDITIONAL ANALYSES FOR SECTION 3.1.

**Additional results.** We report the inference times on the non-convex datasets in Table 2. We report the omitted benchmark results on the small and large convex datasets of Section 3.1 in Figures 5 and 6 and Table 3, which further substantiate our claims. We report the omitted learning curves for `JAXopt` on the large convex and non-convex datasets in Figure 7 and Figure 8, respectively.

**`DC3`: Hyperparameter ablations.** In our benchmarks of Section 3.1, we highlighted two weaknesses in the performance of `DC3`: (i) in both small and large non-convex problems the `RS` is unsatisfactory; (ii) in the large non-convex problem the `CV` is unsatisfactory. We investigated whether these shortcomings can be mitigated by choosing hyperparameters which are different from the defaults. In particular, we tried to address (i) by decreasing the soft penalty parameter for `CV` in `DC3`'s loss function from the default 10.0 to 2.0; (ii) by increasing the number of correction steps from the default 10 to 50. We superimpose the obtained `RS` and `CV` in Figure 9, and report the inference times in Table 4 (only for the case of more correction steps, since changing the soft penalty does not affect inference times). Importantly, these results show both the lack of a clear decoupling between

| Method | Single inference [s] | | | | | Batch inference [s] | | | | |
|---|---|---|---|---|---|---|---|---|---|---|
| | median | LQ | UQ | min | max | median | LQ | UQ | min | max |
| **Non-convex small** | | | | | | | | | | |
| DC3 | 0.0019 | 0.0018 | 0.0019 | 0.0018 | 0.0026 | 0.0020 | 0.0019 | 0.0020 | 0.0019 | 0.0023 |
| Solver | 0.0334 | 0.0298 | 0.0497 | 0.0207 | 0.1213 | 41.748 | 40.200 | 43.198 | 36.764 | 47.149 |
| JAXopt | 0.0134 | 0.0131 | 0.0136 | 0.0122 | 0.0145 | 0.1371 | 0.1364 | 0.1532 | 0.1357 | 0.1540 |
| Πnet | 0.0056 | 0.0055 | 0.0056 | 0.0054 | 0.0072 | 0.0130 | 0.0128 | 0.0131 | 0.0123 | 0.0147 |
| **Non-convex large** | | | | | | | | | | |
| DC3 | 0.0016 | 0.0016 | 0.0017 | 0.0015 | 0.0046 | 0.0248 | 0.0248 | 0.0248 | 0.0247 | 0.0255 |
| Solver | 10.159 | 9.4739 | 10.807 | 7.4350 | 12.612 | 10720 | 9538.4 | 10800 | 9417.8 | 16502 |
| JAXopt | 0.0578 | 0.0575 | 0.0590 | 0.0559 | 0.0596 | 19.430 | 19.430 | 19.430 | 19.429 | 19.430 |
| Πnet | 0.0063 | 0.0063 | 0.0064 | 0.0061 | 0.0092 | 0.2804 | 0.2799 | 0.2807 | 0.2794 | 0.2912 |

Table 2: Inference time comparison for single-instance and batch (1024 contexts) settings, evaluated on the small and large non-convex problems. The table reports the median, lower quartile (LQ, 25th percentile), upper quartile (UQ, 75th percentile), min and max of the runtime.

| Method | Single inference [s] | | | | | Batch inference [s] | | | | |
|---|---|---|---|---|---|---|---|---|---|---|
| | median | LQ | UQ | min | max | median | LQ | UQ | min | max |
| **Convex small** | | | | | | | | | | |
| DC3 | 0.0033 | 0.0032 | 0.0033 | 0.0031 | 0.0050 | 0.0033 | 0.0033 | 0.0034 | 0.0032 | 0.0044 |
| Solver | 0.0019 | 0.0018 | 0.0019 | 0.0011 | 0.0083 | 1.9350 | 1.9264 | 1.9441 | 1.8931 | 2.0282 |
| Solver[†] | 0.0006 | 0.0006 | 0.0006 | 0.0006 | 0.0024 | 0.6190 | 0.6168 | 0.6217 | 0.6100 | 0.6746 |
| JAXopt | 0.0142 | 0.0138 | 0.0150 | 0.0124 | 0.0165 | 0.1603 | 0.1590 | 0.1648 | 0.1581 | 0.1794 |
| Πnet | 0.0055 | 0.0055 | 0.0056 | 0.0053 | 0.0060 | 0.0130 | 0.0128 | 0.0130 | 0.0125 | 0.0136 |
| **Convex large** | | | | | | | | | | |
| DC3 | 0.0072 | 0.0072 | 0.0073 | 0.0071 | 0.0115 | 0.0349 | 0.0349 | 0.0351 | 0.0348 | 0.0369 |
| Solver | 0.6660 | 0.6613 | 0.6716 | 0.3159 | 0.7079 | 680.59 | 676.24 | 682.52 | 662.29 | 687.84 |
| Solver[†] | 0.0603 | 0.0589 | 0.0620 | 0.0466 | 0.9501 | 62.022 | 61.591 | 62.558 | 60.064 | 70.987 |
| JAXopt | 0.0630 | 0.0623 | 0.0641 | 0.0601 | 0.0668 | 21.504 | 21.504 | 21.505 | 21.504 | 21.505 |
| Πnet | 0.0063 | 0.0063 | 0.0064 | 0.0061 | 0.0145 | 0.2800 | 0.2797 | 0.2804 | 0.2794 | 0.2851 |

Table 3: Inference time comparison for single-instance and batch (1024 contexts) settings, evaluated on the small and large convex problems. The table reports the median, lower quartile (LQ, 25th percentile), upper quartile (UQ, 75th percentile), min and max of the runtime. We report results of the Solver (i.e., OSQP) in two modes, normal and parametric labeled with Solver and Solver[†], respectively. Parametric mode means that we inform OSQP that we are repeatedly solving problems with the same structure, which speeds up solution time by reusing calculations across consecutive calls. We note that this is a feature of OSQP that may or may not be available in other solvers.

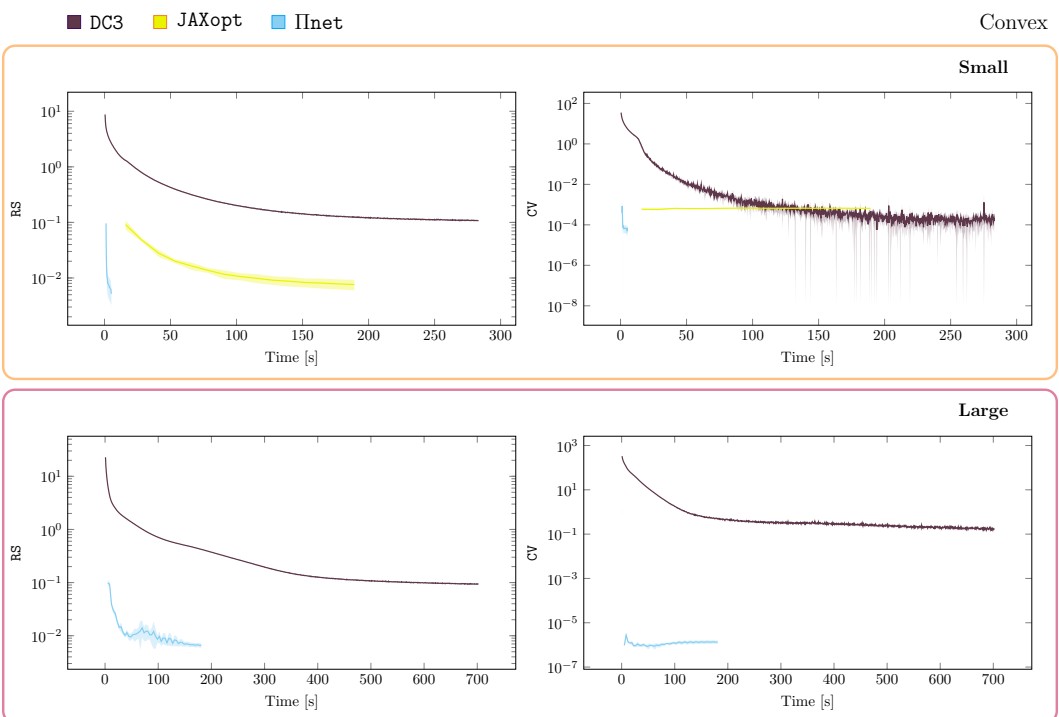

Figure 6: Comparison of the learning curves in terms of average RS and CV on the validation set, on the small and large convex problems. The solid lines denote the mean and the shaded area the standard deviation across 5 seeds. The learning curves for JAXopt on the large dataset are reported in Figure 7 because of the orders of magnitude longer training times.

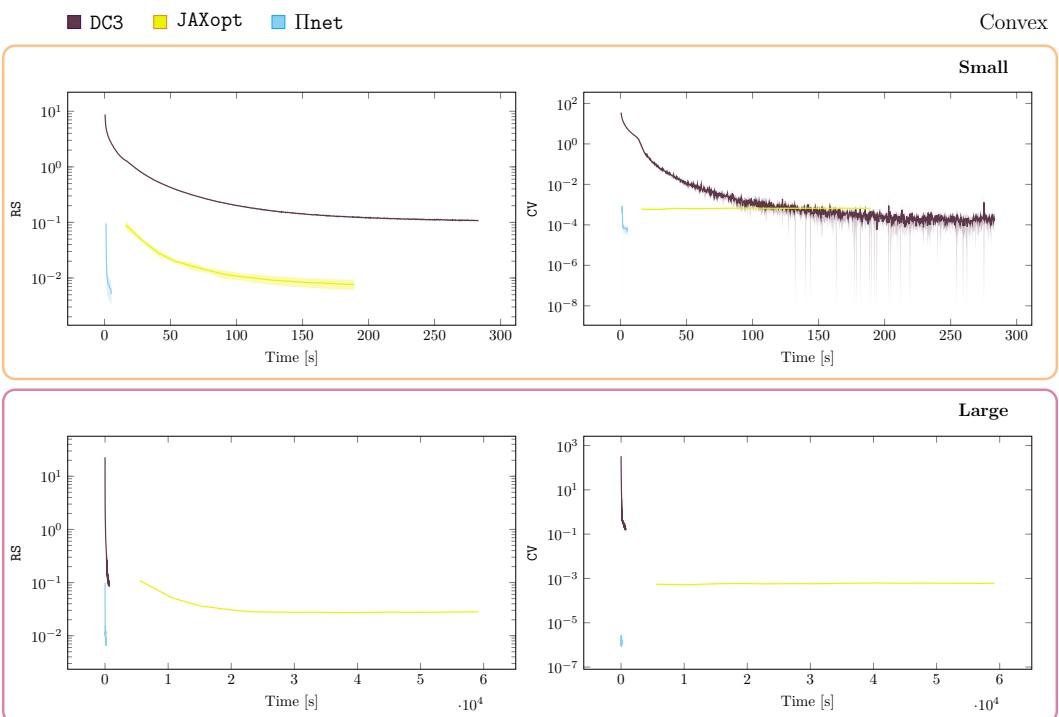

Figure 7: Comparison of the training times in terms of RS and CV for the different methods in the small and large convex problem setting. The solid lines denote the mean and the shaded area the standard deviation across 10 seeds. Note the different timescale on the large dataset results, due to the large training time of JAXopt.

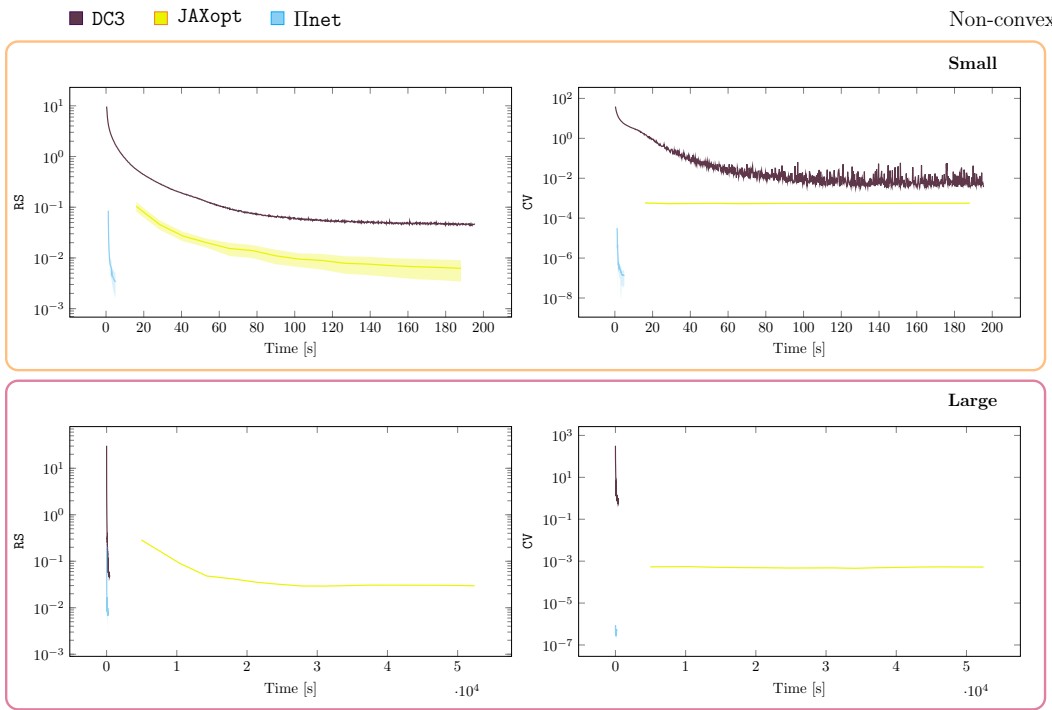

Figure 8: Comparison of the learning curves in terms of average RS and CV on the validation set, on the small and large non-convex problems. The solid lines denote the mean and the shaded area the standard deviation across 5 seeds.

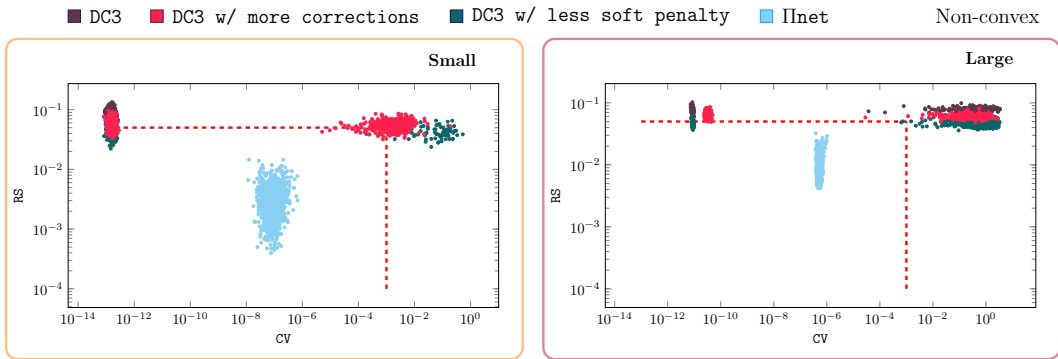

Figure 9: Scatter plots of RS and CV on the small and large non-convex problems on the test set. The red dashed lines show the thresholds to consider a candidate solution optimal. We superimpose the results of the main text with the ones obtained using more correction steps and a smaller soft penalty in the DC3 algorithm.

the various parameters of the DC3 algorithm and the advantages of our projection scheme; see also Appendix C.

### B.3 ADDITIONAL DETAILS AND RESULTS FOR SECTION 3.2

We detail the problem setup for the multi-vehicle motion planning application in Section 3.2 and present additional results. Recall that we denote with $p_i[t] \in \mathbb{R}^m$ the generalized coordinates of vehicle $i$ at the discrete times $t \in \{1, \ldots, T+1\}$, and with $v_i[t]$ and $a_i[t]$ its generalized velocity and acceleration. Its simple discretized dynamics read $v_i[t+1] = v_i[t] + ha_i[t]$, $p_i[t+1] =$

| Method | Single inference [s] | | | | | Batch inference [s] | | | | |
|---|---|---|---|---|---|---|---|---|---|---|
| | median | LQ | UQ | min | max | median | LQ | UQ | min | max |
| **Non-convex small** | | | | | | | | | | |
| DC3 | 0.0080 | 0.0080 | 0.0080 | 0.0079 | 0.0087 | 0.0081 | 0.0081 | 0.0082 | 0.0081 | 0.0083 |
| Solver | 0.9724 | 0.6925 | 1.4587 | 0.3562 | 3.6359 | 1258.8 | 1243.8 | 1264.3 | 1185.2 | 1299.0 |
| Πnet | 0.0056 | 0.0055 | 0.0056 | 0.0054 | 0.0072 | 0.0130 | 0.0128 | 0.0131 | 0.0123 | 0.0147 |
| **Non-convex large** | | | | | | | | | | |
| DC3 | 0.0070 | 0.0069 | 0.0073 | 0.0068 | 0.0105 | 0.1177 | 0.1177 | 0.1177 | 0.1176 | 0.1180 |
| Solver | 6.8153 | 6.7161 | 6.9636 | 6.2371 | 12.046 | 6991.4 | 6986.1 | 7498.7 | 6964.0 | 7523.3 |
| Πnet | 0.0063 | 0.0063 | 0.0064 | 0.0061 | 0.0092 | 0.2804 | 0.2799 | 0.2807 | 0.2794 | 0.2912 |

Table 4: Inference time comparison for single-instance and batch-instance (1024 problems) settings across different methods, evaluated on small and large non-convex problems. The table reports median runtime along with statistical descriptors: lower quartile (LQ, 25th percentile), upper quartile (UQ, 75th percentile), min and max of the runtime. DC3 uses more than the default correction steps.

$p_i[t] + h v_i[t] + \frac{h^2}{2} a_i[t]$. We have constraints on each of these variables and, thus, we consider as optimization variable $y = \begin{bmatrix} p^\top & v^\top & a^\top \end{bmatrix}^\top$, with

$$p = \begin{bmatrix} p_1[0]^\top & \cdots & p_N[0]^\top & \cdots & p_1[T]^\top & \cdots & p_N[T]^\top \end{bmatrix}^\top,$$
$$v = \begin{bmatrix} v_1[0]^\top & \cdots & v_N[0]^\top & \cdots & v_1[T]^\top & \cdots & v_N[T]^\top \end{bmatrix}^\top,$$
$$a = \begin{bmatrix} a_1[0]^\top & \cdots & a_N[0]^\top & \cdots & a_1[T-1]^\top & \cdots & a_N[T-1]^\top \end{bmatrix},$$

where $N$ is the number of vehicles. Below, we use $m = 2$. We want the network to generate trajectories that go from a given set of initial positions $\bar{p}_1[0], \ldots, \bar{p}_N[0]$ to a set of final positions $\bar{p}_1[T+1], \ldots, \bar{p}_N[T+1]$. This is the *context* of the optimization problem, i.e.,

$$\mathbf{x} = \begin{bmatrix} \bar{p}_1[0]^\top & \cdots & \bar{p}_N[0]^\top & \cdots & \bar{p}_1[T+1]^\top & \cdots & \bar{p}_N[T+1]^\top \end{bmatrix}^\top.$$

The constraints on the system are:

- *Dynamic constraints.* The optimization variables $p, v, a$ are related by the system dynamics, via an equality constraint of the type $A_{\text{dyn}} y = 0$.

- *Initial and final positions constraints.* We ensure that the optimal $y$ satisfies the given initial and terminal position constraints via the equality constraint $A_{\text{if}} y = b(\mathbf{x})$. Importantly, we observe that the context affects the constraint via the vector $b(\mathbf{x})$, which in this case is simply $\mathbf{x}$.

- *Workspace, velocity and acceleration constraints.* These constraints impose box constraints $l_p \leq p \leq u_p, l_v \leq v \leq u_v, l_a \leq a \leq u_a$.

- *Jerk constraints.* Jerk constraints limit how abrupt the change in accelerations can be. These are affine inequality constraints of the type $l \leq (a_i[t+1] - a_i[t])/h \leq u$, which we can compactly write as $l_{\text{jerk}} \leq C y \leq u_{\text{jerk}}$ for appropriate $C, l_{\text{jerk}}, u_{\text{jerk}}$.

The objective is to minimize

$$\varphi(y) = \texttt{effort}(y) + \lambda \cdot \texttt{preference}(y) + \nu \cdot \texttt{coverage}(y)$$

where $\texttt{effort}(y)$ describes the input effort of the solution $y$, $\texttt{preference}(y)$ describes the fitness of $y$ with respect to a spatial potential $\psi$, and $\texttt{coverage}(y)$ describes the fraction of the workspace that the agents cover over time, and $\lambda, \nu \geq 0$ are tuning parameters. In our experiment, we use them as binary variables to show the effects of adding certain terms to the objective function. The different terms are defined as follows:

- *Input effort* ($\texttt{effort}$). The input effort is

$$\texttt{effort}(y) = \sum_{i=1}^{N} \sum_{t=0}^{T-1} \|a_i[t]\|^2.$$

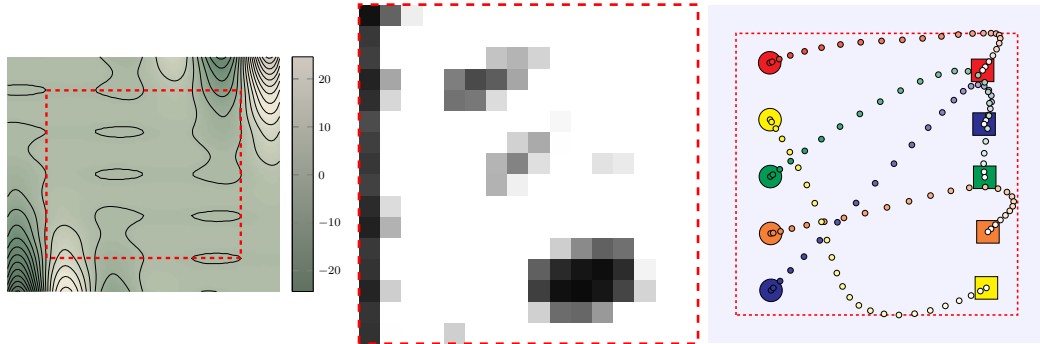

Figure 10: Preference and coverage. On the left we depict the landscape of the potential $\psi(\cdot)$ with the working space delimited by the red, dashed rectangle. In the middle we report the image used to compute the coverage score of the trajectory on the right.

- *Individual contribution* (`preference`). Each vehicle tries to minimize the cumulative value along the path of the scaled Ishigami potential (Terpin et al., 2024b)

$$\psi\left(\frac{z}{1.25}\right) = 0.05\left(\sin(z_1) + 7\sin(z_2)^2 + \frac{1}{10}\left(\frac{z_1 + z_2}{2}\right)^4 \sin(z_1)\right),$$

depicted on the left of Figure 10. That is,

$$\texttt{preference}(y) = \sum_{i=1}^{N}\sum_{t=0}^{T}\psi(p_i[t]).$$

- *Fleet contribution* (`coverage`). We define the coverage over time as the fraction of the space $[(p_{\min})_1, (p_{\max})_1] \times [(p_{\min})_2, (p_{\max})_2]$ the vehicles sweep over during their trajectory. We map the position of the $i^{\text{th}}$ vehicle into a continuous pixel-space of size $H \times W = 16 \times 16$ via

$$u_i = \frac{(p_i[t])_1 - (p_{\min})_1}{(p_{\max})_1 - (p_{\min})_1}W, \qquad v_i = H - \frac{(p_i[t])_2 - (p_{\min})_2}{(p_{\max})_2 - (p_{\min})_2}H.$$

On each pixel $(u, v)$ we place a bivariate Gaussian

$$G_i(u, v) = A\exp\left(-\frac{1}{2(1-\rho^2)}\left[\frac{(u-u_i)^2}{\sigma_x^2} + \frac{(v-v_i)^2}{\sigma_y^2} - \frac{2\rho(u-u_i)(v-v_i)}{\sigma_x\sigma_y}\right]\right),$$

where $A = 200, \sigma_x = \sigma_y = 1, \rho = 0$ are hyperparameters. We sum (and clip) the $N$ kernels to obtain a coverage image,

$$I(u, v) = \min\left(\sum_{i=1}^{N}G_i(u, v), 255\right).$$

We display $I$ for a sampled trajectory in Figure 10. Finally, the coverage score is computed as

$$\texttt{coverage}(y) = -\frac{1}{HW}\sum_{p=1}^{H}\sum_{q=1}^{W}\frac{I(p, q)}{255},$$

measuring the fraction of pixels covered by the Gaussian footprints in a differentiable manner.

Overall, the optimization problem reads:

$$\underset{y\in\mathbb{R}^d}{\text{minimize}} \quad \varphi(y) \quad \text{subject to} \quad \begin{bmatrix} A_{\text{if}} \\ A_{\text{dyn}} \end{bmatrix}y = \begin{bmatrix} b(\text{x}) \\ 0 \end{bmatrix}, \quad \begin{bmatrix} l_p \\ l_v \\ l_a \end{bmatrix} \leq y \leq \begin{bmatrix} u_p \\ u_v \\ u_a \end{bmatrix}, \quad l_{\text{jerk}} \leq Cy \leq u_{\text{jerk}}.$$

In Figures 11 and 12 we report various trajectories generated with $\Pi\texttt{net}$ with different number of vehicles $N \in \{5, 15\}$ and horizon $T = 25$. These additional qualitative results show the effectiveness of $\Pi\texttt{net}$ in synthesizing trajectories that optimize non-convex, fleet-level preferences also for large fleets and long horizons: in the largest setting reported here, there are $n = 3030$ optimization variables ($d = 2280$).

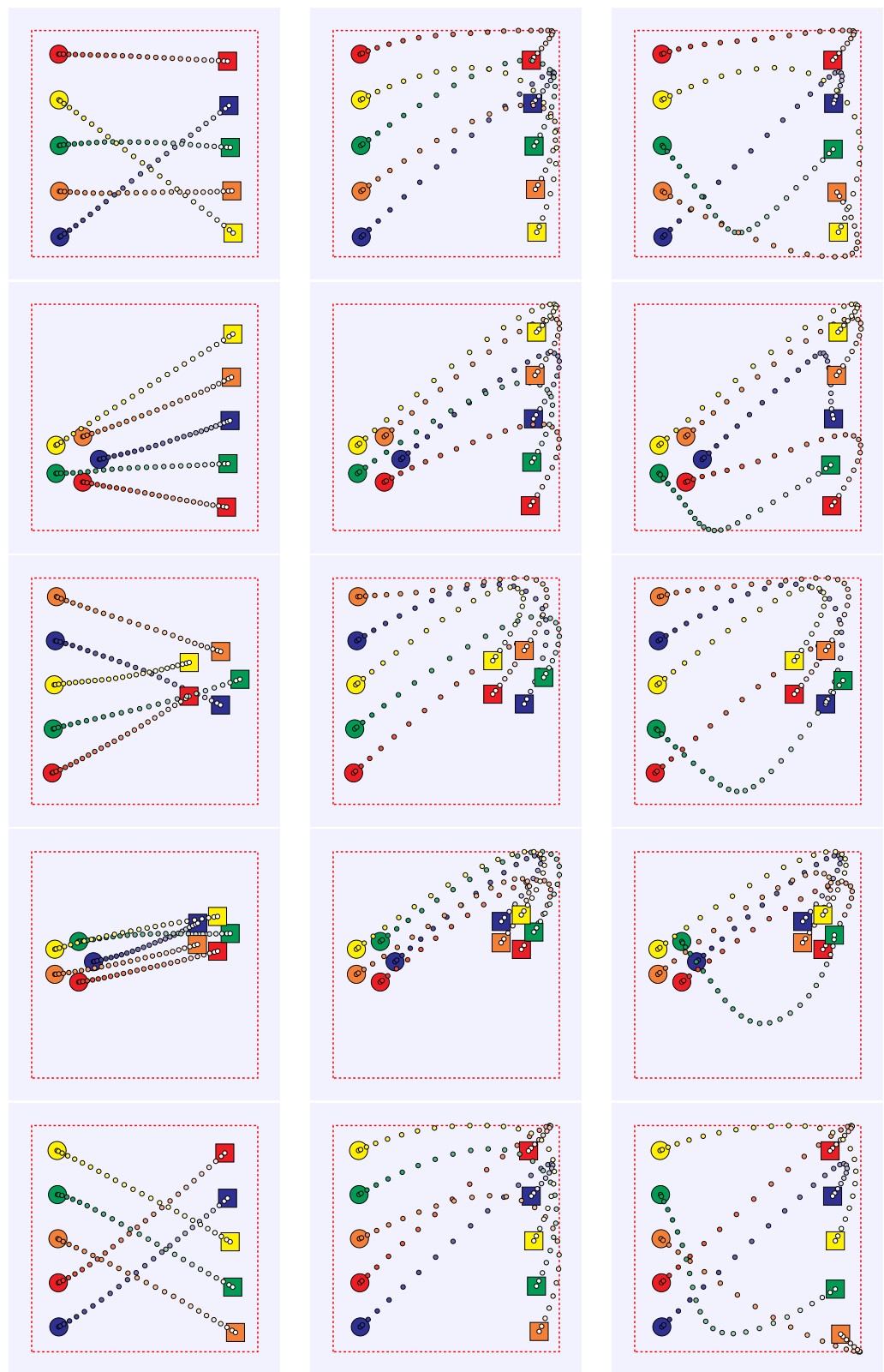

Figure 11: Collection of synthesized 5-vehicles trajectories. Each row relates to a different instance of initial and final configurations, and from left to right we report the generated trajectory with a network trained with $(\lambda, \nu) = (0, 0), (\lambda, \nu) = (1, 0)$ and $(\lambda, \nu) = (1, 1)$, respectively.

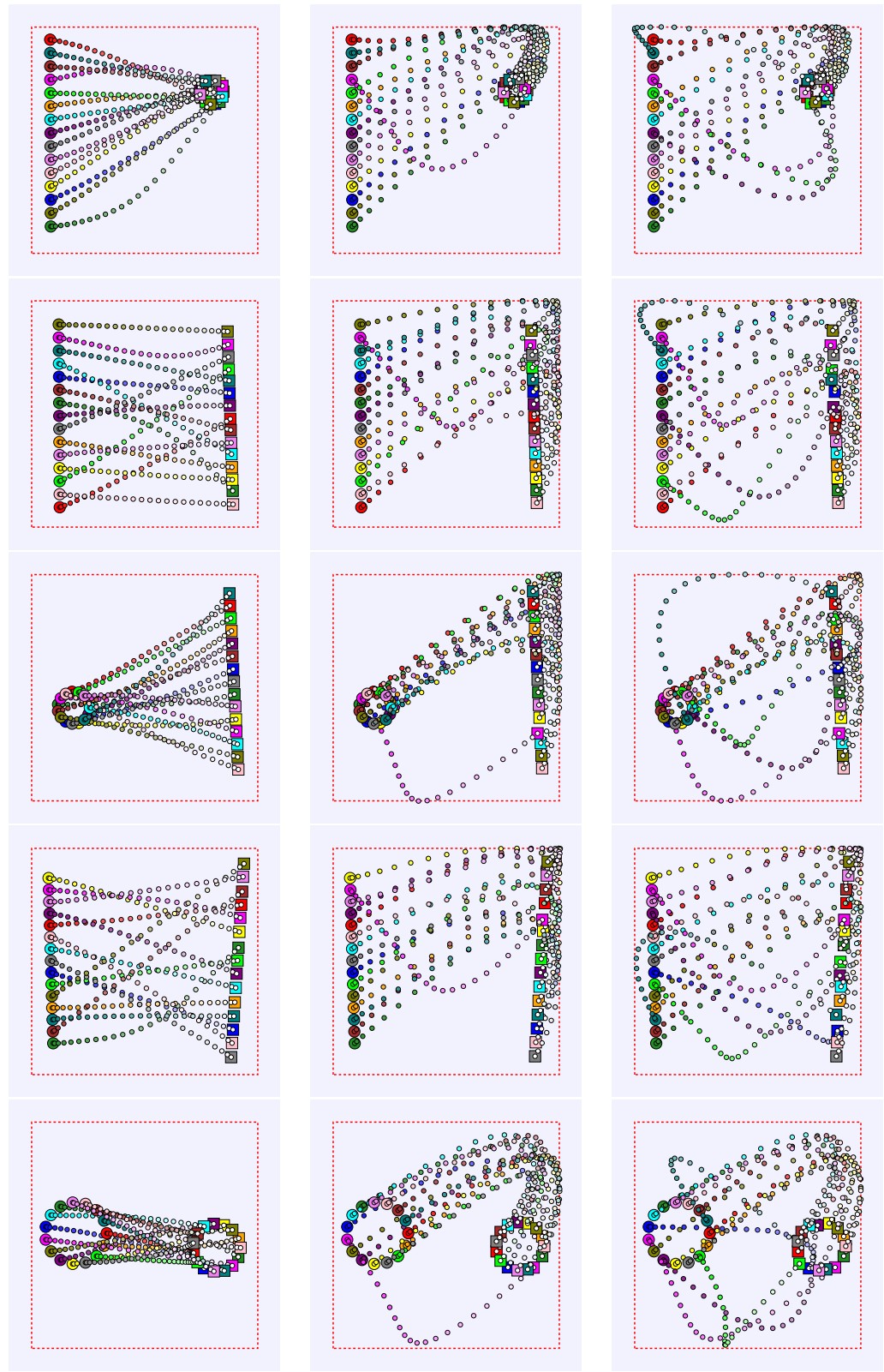

Figure 12: Collection of synthesized 15-vehicles trajectories. Each row relates to a different instance of initial and final configurations, and from left to right we report the generated trajectory with a network trained with $(\lambda, \nu) = (0, 0)$, $(\lambda, \nu) = (1, 0)$ and $(\lambda, \nu) = (1, 1)$, respectively.

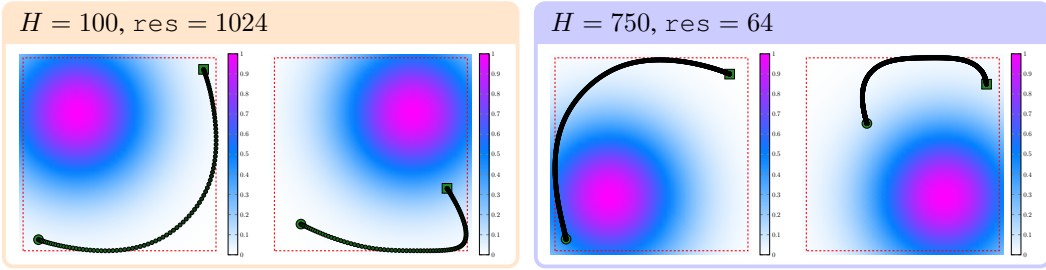

Figure 13: Examples of resulting trajectories for the experiments in Appendix B.4.

### B.4 Contextual coverage and long-horizon planning

Building on top of the trajectory planning application, we provide some preliminary evidence that Πnet is capable of handling:

1. very high-dimensional context size (in the millions of variables); and
2. a large number of optimization variables and constraints (in the tens of thousands).

For this, we focus on a single vehicle with the same constraints as in the previous section, increase the horizon length, and consider as cost function the average along the trajectory of the value assigned (via bilinear interpolation) to the position of the vehicle by a discrete map m of resolution res × res. Thus, the context x corresponds to the initial and final location as well as the map m. We generate the map m by sampling uniformly at random one of four locations

$$c \in \{(-2.5, -2.5), (-2.5, 2.5), (2.5, -2.5), (2.5, 2.5)\}.$$

Then, the res × res pixels comprising the map m correspond to a uniform discretization of the grid $[-5, 5] \times [-5, 5]$, with values resulting from a Gaussian with center $c$ and a standard deviation of 2. This simple construction (in the sense that the single parameter controlling the map m is which $c$ was used) allows to test how Πnet deals with backbone networks that have as input high-dimensional contexts, while keeping fixed the number of optimization variables.

For this, we use a convolutional neural network with three layers, 16 features and downsampling factor $D$ for each layer, followed by a linear layer and a softmax, resulting in 4 logits. These are appended to the vector input (the initial and final positions) as input to the same MLP used in the previous experiments. We note that the convolutional neural network and the MLP are trained jointly as the backbone of Πnet. We consider two cases:

1. $H = 100$, res $= 1024$, $D = 4$. In this case, we evaluate if Πnet enables training end-to-end of neural networks with constraints even when the context is very high-dimensional (for a comparison, Donti et al. (2021) consider a context of dimension 50).

2. $H = 750$, res $= 64$, $D = 2$. In this case, we evaluate Πnet for a context of size similar to that typical of, e.g., reinforcement learning settings and a number of optimization variables and constraints two orders of magnitude larger than state-of-the-art parametric optimization algorithms (for instance, Donti et al. (2021) consider 100 optimization variables, and 50 equality and inequality constraints).

In both settings, Πnet successfully learns to solve the planning task, as qualitatively shown in Figure 13. Despite the simplicity of the maps used, we believe that these examples show the promise of using Πnet for high-dimensional optimization problems with very large contexts.

### B.5 Second order cone constraints.

In this section, we provide evidence that Πnet can be used to solve parametric problems with second-order cone constraints. By doing so, we also elaborate on how one can introduce additional constraint types into Πnet effortlessly.

**Experimental setup.** We focus on problems of the form:

$$\underset{y=(y_1,y_2)}{\text{minimize}} \quad c^\top y_1 \quad \text{subject to} \quad Ay_1 + y_2 = b, \ y_2 \in \mathcal{K}'_1, \tag{11}$$

where $A \in \mathbb{R}^{d_2 \times d_1}, c, y_1 \in \mathbb{R}^{d_1}, y_2, b \in \mathbb{R}^{d_2}, \mathcal{K}_1' = \{y_2 \in \mathbb{R}^{d_2} \mid \|y_{2,:-1}\|_2 \le y_{2,-1}\}$ is a second-order cone, and $y_{2,:-1}$ denotes the first $d_2 - 1$ entries of $y_2$, and $y_{2,-1}$ the last one. We make the following two observations:

- First, the feasible set is non-linear and possibly unbounded. Thus, methods like GLinSAT (Zeng et al., 2024) are not applicable.

- Second, one cannot train an unconstrained network and then apply the projection layer, since the optimization objective is linear and, thus, the network would diverge. See also Appendix C.4.

To generate the problem data, we follow the procedure described by O'donoghue et al. (2016, Section 6.6), reported here for completeness and to clarify the elements of the random parametric problems we consider. First, we generate a random matrix $A \in \mathbb{R}^{d_2 \times d_1}$ sampling uniformly at random in $[-1, 1]^{d_2 \times d_1}$. The matrix $A$ is fixed for all the problems of a training run, i.e., it is not a context/parameter of the problem. In other words, $A$ is context-independent. Given $A$, we generate a batch of $B$ problems as follows:

1. We sample $B$ random vectors $z$ uniformly in $[-1, 1]^{d_2}$ and $B$ random primal solutions $y_1^\star \in \mathbb{R}^{d_1}$.

2. We project $z$ onto the second-order cone to obtain $y_2^\star = \Pi_{\mathcal{K}_1'}(z)$.

3. We set $b = Ay_1^\star + y_2^\star$ and $c = -A^\top(y_2^\star - z)$.

The optimal value of the problem is then given by $c^\top y_1^\star$, and the context x (i.e., the input to the NN), is x $= (b, c)$. For $\Pi$net, we employ the same MLP used in Appendix A followed by the $\Pi$net projection layer, so that the optimal output should yield $\hat{y} = (\hat{y}_1, \hat{y}_2)$ with $c^\top \hat{y}_1 \approx c^\top y_1^\star$. For the experiments in this section, we use $(d_1, d_2) \in \{(25, 25), (500, 500)\}$ (we refer to these configurations as small and large, respectively) and $B = 128$. We train the network for 1000 epochs with a different batch at each epoch, and then generate an additional one for evaluation at the end of the training, on which we calculate the statistics. We compare the performance of our method against:

- `cvxpylayers` Agrawal et al. (2019), which we use as a replacement of our custom projection layer. This is the same procedure we adopted in Section 3.1 for `JAXopt`. Here, we consider `cvxpylayers` because `JAXopt` does not support second-order cone constraints.

- `SCS` O'donoghue et al. (2016), a traditional first-order solver for second-order cone programs. To make our comparison with `SCS` as fair and comprehensive as possible, we perform two benchmarks. On one, we interface `SCS` through the very popular parser `CVXPY` (Diamond & Boyd, 2016) using its Disciplined Parametrized Programming functionality, as many users would do. On the other benchmark, we interface `SCS` directly and exploited its native parametric programming functionality, as more advanced users would.

**Remark.** *It is important to note that with these benchmarks we are exploiting the full parametric capabilities of existing solvers, which is not always the case by end-users or even other hard-constrained NN benchmarks.*

Similarly to the linearly-constrained benchmarks in Section 3.1, we consider learning curves, RS, CV, and inference times.

**Integrating (11) into $\Pi$net.** Equivalently, we can write (11) as:

$$\underset{y}{\text{minimize}} \quad \begin{bmatrix} c^\top & 0 \end{bmatrix} y \quad \text{subject to} \quad \begin{bmatrix} A & I \end{bmatrix} y = b, \quad y \in \mathcal{K}_1, \tag{12}$$

with $\mathcal{K}_1 = \mathbb{R}^n \times \mathcal{K}_1'$, $\mathcal{K}_1'$ being the second-order cone, and without $\mathcal{K}_2$ since we do not use any auxiliary variable (i.e., $n = d = d_1 + d_2$). We recall here that $\mathcal{K}_1$ and $\mathcal{K}_2$ refer to the problem formulation in Section 2.1. In view of Appendix E, this amounts to a lifted formulation of second-order cones and is thus without loss of generality. Now, (12) is in our standard formulation and we can apply Algorithm 1. In particular, with the notation used in Algorithm 1, we only need to define the projection

$$t_{k+1} = \Pi_{\mathcal{K}_1} \left( \frac{2z_{k+1,1} - s_{k,1} + 2\sigma y_{\text{raw},1}}{1 + 2\sigma} \right).$$

Denoting with $t_{k+1,:d_1}, z_{k+1,1,:d_1}, s_{k,1,:d_1}, y_{\text{raw},:d_1}$ the first $d_1$ entries of the vectors $t_{k+1}, z_{k+1,1}, s_{k,1}, y_{\text{raw},1}$ and with $t_{k+1,d_1:}, z_{k+1,1,d_1:}, s_{k,1,d_1:}, y_{\text{raw},d_1:}$ the remaining $d_2$ entries, we

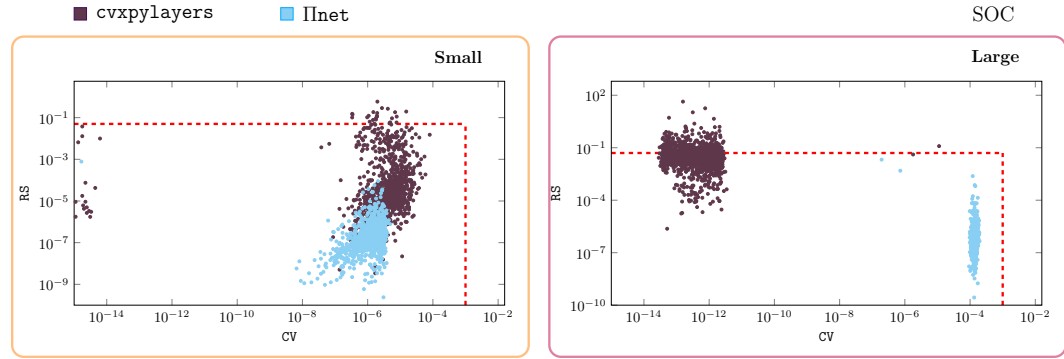

Figure 14: Scatter plots of RS and CV on the second-order cone programs on the test set. The red dashed lines show the thresholds to consider a candidate solution optimal.

| Method | Single inference [s] | | | | | Batch inference [s] | | | | |
|---|---|---|---|---|---|---|---|---|---|---|
| | median | LQ | UQ | min | max | median | LQ | UQ | min | max |
| **Small second-order cone programs** | | | | | | | | | | |
| CVXPY | 0.00076 | 0.00074 | 0.00078 | 0.00071 | 0.00084 | 0.59085 | 0.58784 | 0.59399 | 0.57668 | 0.70665 |
| SCS | 0.00005 | 0.00004 | 0.00005 | 0.00004 | 0.00005 | 0.04612 | 0.04580 | 0.04652 | 0.04483 | 0.04926 |
| cvxpylayers | 0.00857 | 0.00813 | 0.01270 | 0.00706 | 0.06567 | 1.82553 | 1.68570 | 4.47413 | 1.65853 | 5.73859 |
| Πnet | 0.00905 | 0.00884 | 0.00928 | 0.00848 | 0.01137 | 0.02265 | 0.02245 | 0.02292 | 0.02202 | 0.02415 |
| **Large second-order cone programs** | | | | | | | | | | |
| CVXPY | 0.06721 | 0.06658 | 0.06743 | 0.06578 | 0.06899 | 62.70289 | 62.65441 | 62.76826 | 62.46146 | 63.31168 |
| SCS | 0.01072 | 0.01049 | 0.01078 | 0.00976 | 0.01155 | 10.97265 | 10.80431 | 11.10839 | 10.68313 | 26.18852 |
| cvxpylayers | 2.36724 | 2.29826 | 2.41810 | 2.18466 | 4.04885 | 342.35211 | 341.85465 | 343.06660 | 341.65177 | 344.11561 |
| Πnet | 0.01219 | 0.01187 | 0.01356 | 0.01113 | 0.04383 | 0.90982 | 0.90932 | 0.91134 | 0.90844 | 0.94499 |

Table 5: Inference time comparison for single-instance and batch-instance (1024 problems) settings across different methods, evaluated on the second-order cone programs. The table reports median runtime along with statistical descriptors: lower quartile (LQ, 25th percentile), upper quartile (UQ, 75th percentile), min and max of the runtime.

have

$$t_{k+1,:d_1} = \frac{2z_{k+1,1,:d_1} - s_{k,1,:d_1} + 2\sigma y_{\mathrm{raw},1,:d_1}}{1 + 2\sigma} \quad \text{and}$$

$$t_{k+1,d_1:} = \Pi_{\mathcal{K}'_1}\left(\frac{2z_{k+1,2,d_1:} - s_{k,2,d_1:} + 2\sigma y_{\mathrm{raw},2,d_1:}}{1 + 2\sigma}\right).$$

The projection $\Pi_{\mathcal{K}'_1}(\cdot)$ onto the second-order cone admits a closed form expression; see, e.g., the work of Busseti et al. (2019).

**Results.** In Figure 14 we report the RS and CV for each problem instance in the test set. Analogously to the linearly constrained benchmarks in Section 3.1, we consider a candidate solution to be optimal if the condition $CV \leq 10^{-3}$ and $RS \leq 5\%$ is satisfied. We report training curves in Figure 15 and inference times in Table 5.

Compared to cvxpylayers, Πnet can train and perform inference significantly faster, even orders of magnitude for large problems. Moreover, Πnet provides solutions with both lower RS and CV. Compared to SCS, Πnet is faster at inference for a batch of small problems, and both a single and a batch of large problems. Instead, SCS is faster for single small problems.

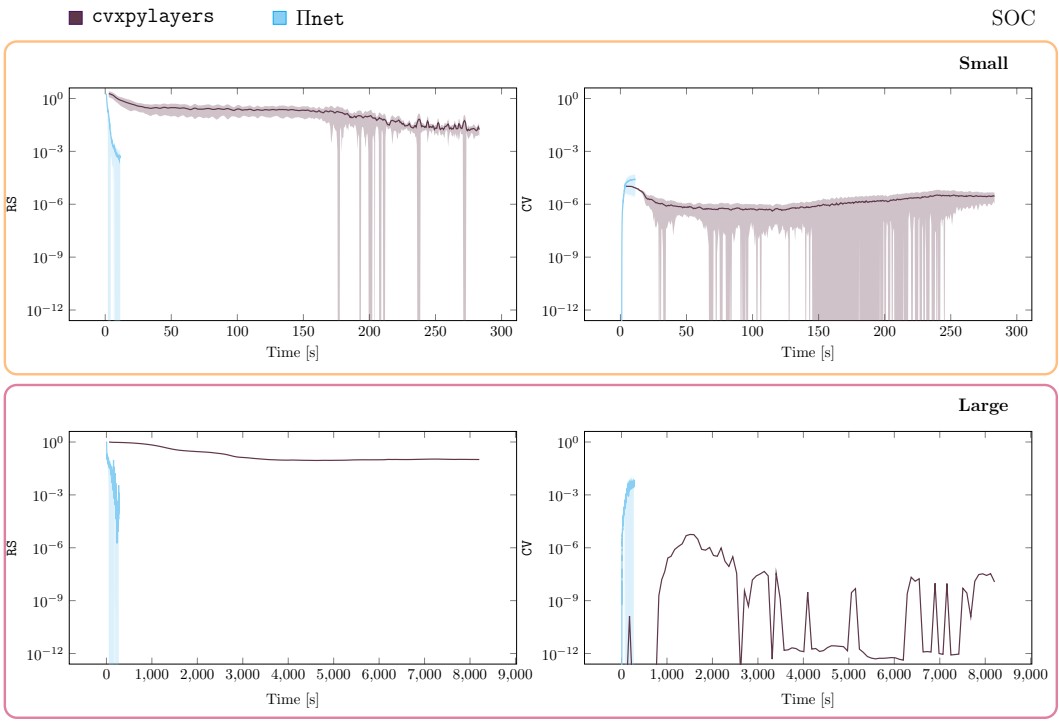

Figure 15: Comparison of the learning curves in terms of average RS and CV on the validation set, on the second-order cone programs. The solid lines denote the mean and the shaded area the standard deviation across 5 seeds.

## C    THE SHARP BITS

Numerical optimization is an art of details, and this section describes the sharp bits of $\Pi$net, which are fundamental to make HCNNs work reliably.

### C.1    MATRIX EQUILIBRATION

The precision and convergence rates of both the forward (Section 2.1.1) and backward (Section 2.1.2) passes in our projection layer are heavily influenced, via (4a) and (9), by the *condition number* of the matrix $A(\mathrm{x})$ in the equality constraint $A(\mathrm{x}) \begin{bmatrix} y \\ y_{\text{aux}} \end{bmatrix} = b(\mathrm{x})$ (Wathen, 2015). The condition number of a matrix $A$ (we drop the dependency on x for clarity of exposition) is defined as the ratio of its maximum singular value to its minimum singular value, and the procedure of *decreasing* this ratio for improving numerical performance is called *preconditioning* or *equilibration*.

To achieve this, we seek a *diagonal scaling* (Wathen, 2015). The idea is that to solve $A \begin{bmatrix} y \\ y_{\text{aux}} \end{bmatrix} = b$, one can instead solve $D_r A D_c \begin{bmatrix} \tilde{y} \\ \tilde{y}_{\text{aux}} \end{bmatrix} = D_r b$ for invertible matrices $D_r \in \mathbb{R}^{m \times m}, D_c \in \mathbb{R}^{n \times n}$ and recover the solution as $\begin{bmatrix} y \\ y_{\text{aux}} \end{bmatrix} = D_c \begin{bmatrix} \tilde{y} \\ \tilde{y}_{\text{aux}} \end{bmatrix}$. We implement a modified version of Ruiz's equilibration (Wathen, 2015) which is outlined in Algorithm 4. In our numerical experiments we use $K = 25, \varepsilon = 10^{-3}$ and the Gauss-Seidel update mode.

---

ALGORITHM 4. Modified Ruiz's equilibration

---

**Inputs**: $A \in \mathbb{R}^{m \times n}$, maximum iterations $K$, tolerance $\varepsilon$, update mode (Gauss-Seidel or Jacobi)
$D_r \leftarrow I_m, D_c \leftarrow I_n$
$A_{\text{scaled}} \leftarrow A$
**For** $k = 1$ **to** $K$**:**
  **If Gauss-Seidel update:**
    Compute row norms $d_{r,i} = \|(A_{\text{scaled}})_{i,:}\|_2$ for all $i$
    $D_r \leftarrow \text{diag}(1/\sqrt{d_{r,1}}, \ldots, 1/\sqrt{d_{r,m}}) \cdot D_r$
    $A_{\text{scaled}} \leftarrow \text{diag}(1/\sqrt{d_{r,1}}, \ldots, 1/\sqrt{d_{r,m}}) \cdot A_{\text{scaled}}$
    Compute column norms $d_{c,i} = \|(A_{\text{scaled}})_{:,i}\|_2$ for all $i$
    $D_c \leftarrow D_c \cdot \text{diag}(1/\sqrt{d_{c,1}}, \ldots, 1/\sqrt{d_{c,n}})$
    $A_{\text{scaled}} \leftarrow A_{\text{scaled}} \cdot \text{diag}(1/\sqrt{d_{c,1}}, \ldots, 1/\sqrt{d_{c,n}})$
  **Else:**
    Compute row norms $d_{r,i} = \|(A_{\text{scaled}})_{i,:}\|_2$ for all $i$
    Compute column norms $d_{c,i} = \|(A_{\text{scaled}})_{:,i}\|_2$ for all $i$
    $D_r \leftarrow \text{diag}(1/\sqrt{d_{r,1}}, \ldots, 1/\sqrt{d_{r,m}}) \cdot D_r$
    $D_c \leftarrow D_c \cdot \text{diag}(1/\sqrt{d_{c,1}}, \ldots, 1/\sqrt{d_{c,n}})$
    $A_{\text{scaled}} \leftarrow \text{diag}(1/\sqrt{d_{r,1}}, \ldots, 1/\sqrt{d_{r,m}}) \cdot A_{\text{scaled}} \cdot \text{diag}(1/\sqrt{d_{c,1}}, \ldots, 1/\sqrt{d_{c,n}})$
  Compute row norms $d_{r,i} = \|(A_{\text{scaled}})_{i,:}\|_2$ for all $i$
  Compute column norms $d_{c,i} = \|(A_{\text{scaled}})_{:,i}\|_2$ for all $i$
  **If** $(1 - \min(d_{r,i})/\max(d_{r,i})) < \varepsilon$ and $(1 - \min(d_{c,i})/\max(d_{c,i})) < \varepsilon$**:**
    **Return** $D_r, D_c$
**Return** $D_r, D_c$

---

Next, we describe how the matrix equilibration affects the Douglas-Rachford algorithm in the case of polytopic constraint sets. To start, we observe that $\mathcal{K}$ is a box constraint and, thus, can be written as

$$\mathcal{K} = \mathcal{K}_1 \times \mathcal{K}_2 = \mathcal{K}_{1,1} \times \ldots \times \mathcal{K}_{1,d} \times \mathcal{K}_{2,1} \times \ldots \times \mathcal{K}_{2,n-d}.$$

Then, we rewrite (3) in terms of the new coordinates:

$$\begin{bmatrix} \tilde{y} \\ \tilde{y}_{\text{aux}} \end{bmatrix} = D_c^{-1} \begin{bmatrix} y \\ y_{\text{aux}} \end{bmatrix} = \begin{bmatrix} D_{c,1}^{-1} & \\ & D_{c,2}^{-1} \end{bmatrix} \begin{bmatrix} y \\ y_{\text{aux}} \end{bmatrix}.$$

That is, since

$$\begin{bmatrix} y \\ y_{\text{aux}} \end{bmatrix} \in \mathcal{A} \iff \begin{bmatrix} \tilde{y} \\ \tilde{y}_{\text{aux}} \end{bmatrix} \in \tilde{\mathcal{A}} = \{v \mid D_r A D_c v = D_r b\} = \left\{ v \mid \tilde{A} v = \tilde{b} \right\}$$

$$\begin{bmatrix} y \\ y_{\text{aux}} \end{bmatrix} \in \mathcal{K} \iff \begin{bmatrix} \tilde{y} \\ \tilde{y}_{\text{aux}} \end{bmatrix} \in \tilde{\mathcal{K}} = d_{c,1}^{-1} \mathcal{K}_{1,1} \times \ldots \times d_{c,d}^{-1} \mathcal{K}_{1,d}$$

$$\times d_{c,d+1}^{-1} \mathcal{K}_{2,1} \times \ldots \times d_{c,n}^{-1} \mathcal{K}_{2,n-d}$$

$$= \tilde{\mathcal{K}}_{1,1} \times \ldots \times \tilde{\mathcal{K}}_{1,d} \times \tilde{\mathcal{K}}_{2,1} \times \ldots \times \tilde{\mathcal{K}}_{2,n-d}$$

$$= \tilde{\mathcal{K}}_1 \times \tilde{\mathcal{K}}_2,$$

we seek to solve

$$(\Pi_{\mathcal{C}}(y_{\text{raw}}), y_{\text{aux}}^\star) = D_c \cdot \operatorname*{argmin}_{\tilde{y}, \tilde{y}_{\text{aux}}} \left\{ \|D_{c,1}\tilde{y} - y_{\text{raw}}\|^2 + \mathcal{I}_{\tilde{\mathcal{A}}} \left( \begin{bmatrix} \tilde{y} \\ \tilde{y}_{\text{aux}} \end{bmatrix} \right) + \mathcal{I}_{\tilde{\mathcal{K}}} \left( \begin{bmatrix} \tilde{y} \\ \tilde{y}_{\text{aux}} \end{bmatrix} \right) \right\}. \qquad (13)$$

Then, we split the objective function as follows

$$\tilde{g}\left( \begin{bmatrix} \tilde{y} \\ \tilde{y}_{\text{aux}} \end{bmatrix} \right) = \mathcal{I}_{\tilde{\mathcal{A}}} \left( \begin{bmatrix} \tilde{y} \\ \tilde{y}_{\text{aux}} \end{bmatrix} \right) \quad \text{and} \quad \tilde{h}\left( \begin{bmatrix} \tilde{y} \\ \tilde{y}_{\text{aux}} \end{bmatrix} \right) = \|D_{c,1}\tilde{y} - y_{\text{raw}}\|^2 + \mathcal{I}_{\tilde{\mathcal{K}}} \left( \begin{bmatrix} \tilde{y} \\ \tilde{y}_{\text{aux}} \end{bmatrix} \right)$$

and by applying the Douglas-Rachford algorithm we obtain the fixed-point iteration

$$k = 0, 1, \ldots, K-1 \quad \left| \begin{aligned} \tilde{z}_{k+1} &= \begin{bmatrix} \tilde{z}_{k,1} \\ \tilde{z}_{k,2} \end{bmatrix} = \operatorname{prox}_{\sigma \tilde{g}}(s_k) = \Pi_{\tilde{\mathcal{A}}}(\tilde{s}_k) & (14a) \\[2mm] \tilde{t}_{k+1} &= \operatorname{prox}_{\sigma \tilde{h}}(2\tilde{z}_{k+1} - s_k) = \begin{bmatrix} \diamondsuit \\ \Pi_{\tilde{\mathcal{K}}_2}(2\tilde{z}_{k+1,2} - \tilde{s}_{k,2}) \end{bmatrix} & (14b) \\[2mm] \tilde{s}_{k+1} &= \begin{bmatrix} \tilde{s}_{k+1,1} \\ \tilde{s}_{k+1,2} \end{bmatrix} = \tilde{s}_k + \omega(\tilde{t}_{k+1} - \tilde{z}_{k+1}) & (14c) \end{aligned} \right.$$

where $\diamondsuit = [\ldots \quad \diamondsuit_i \quad \ldots]^\top$, with (using the notation $(v)_i$ to indicate the $i^{\text{th}}$ entry of the vector $v$)

$$\diamondsuit_i = \Pi_{\tilde{\mathcal{K}}_{1,i}} \left( \frac{2\sigma d_{c,i} y_{\text{raw}} + 2(\tilde{z}_{k+1,1})_i - (\tilde{s}_{k,1})_i}{1 + 2\sigma d_{c,i}^2} \right).$$

The equilibration effectively only changes (14b). To prove the update in (14b), we note that equilibration does not affect the solution of the proximal in (14b) with respect to the auxiliary variables:

$$\begin{aligned} \operatorname{prox}_{\sigma \tilde{h}}(2\tilde{z}_{k+1} - \tilde{s}_k) &= \operatorname*{argmin}_{\tilde{v}, \tilde{v}_{\text{aux}}} \|D_{c,1}\tilde{v} - y_{\text{raw}}\|^2 + \mathcal{I}_{\tilde{\mathcal{K}}} \left( \begin{bmatrix} \tilde{v} \\ \tilde{v}_{\text{aux}} \end{bmatrix} \right) + \frac{1}{2\sigma} \left\| \begin{bmatrix} \tilde{v} \\ \tilde{v}_{\text{aux}} \end{bmatrix} - 2\tilde{z}_{k+1} + \tilde{s}_k \right\|^2 \\ &= \begin{bmatrix} \operatorname*{argmin}_{\tilde{v} \in \tilde{\mathcal{K}}_1} \|D_{c,1}\tilde{v} - y_{\text{raw}}\|^2 + \frac{1}{2\sigma}\|\tilde{v} - 2\tilde{z}_{k+1,1} + \tilde{s}_{k,1}\|^2 \\ \operatorname*{argmin}_{\tilde{v}_{\text{aux}} \in \tilde{\mathcal{K}}_2} \frac{1}{2\sigma}\|\tilde{v}_{\text{aux}} - 2\tilde{z}_{k+1,2} + \tilde{s}_{k,2}\|^2 \end{bmatrix} \\ &= \begin{bmatrix} \diamondsuit \\ \Pi_{\tilde{\mathcal{K}}_2}(2\tilde{z}_{k+1,2} - \tilde{s}_{k,2}) \end{bmatrix}. \end{aligned}$$

In particular, since

$$l \le y_{\text{aux}} \le u \iff D_{c,2}^{-1}l \le D_{c,2}^{-1}y_{\text{aux}} \le D_{c,2}^{-1}u \iff D_{c,2}^{-1}l \le \tilde{y}_{\text{aux}} \le D_{c,2}^{-1}u$$

the projection onto $\tilde{\mathcal{K}}_2$ remains a projection onto a box, but we modify the upper and lower bounds of the box according to $D_{c,2}$. We can write $\diamondsuit$, by completing the square, as

$$\operatorname*{argmin}_{\tilde{v} \in \tilde{\mathcal{K}}_1} \left\| \left( I_d + 2\sigma D_{c,1}^2 \right) \tilde{v} - (2\sigma D_{c,1} y_{\text{raw}} + 2\tilde{z}_{k+1,1} - \tilde{s}_{k,1}) \right\|^2$$

and, thus,

$$\diamond_i = \operatorname*{argmin}_{\tilde{v} \in \tilde{\mathcal{K}}_{1,i}} \left\| \left(1 + 2\sigma d_{c,i}^2\right) \tilde{v} - \left(2\sigma d_{c,i} y_{\mathrm{raw}} + 2(\tilde{z}_{k+1,1})_i - (\tilde{s}_{k,1})_i\right) \right\|^2$$

$$= \operatorname*{argmin}_{\tilde{v} \in \tilde{\mathcal{K}}_{1,i}} \left\| \tilde{v} - \frac{2\sigma d_{c,i} y_{\mathrm{raw}} + 2(\tilde{z}_{k+1,1})_i - (\tilde{s}_{k,1})_i}{1 + 2\sigma d_{c,i}^2} \right\|^2$$

$$= \Pi_{\tilde{\mathcal{K}}_{1,i}} \left( \frac{2\sigma d_{c,i} y_{\mathrm{raw}} + 2(\tilde{z}_{k+1,1})_i - (\tilde{s}_{k,1})_i}{1 + 2\sigma d_{c,i}^2} \right).$$

Although we described the case with box constraints, our changes can be adapted for other constraint classes such as second-order cone constraints by equilibrating equally variables that are coupled by the constraints. We leave these modifications for future work.

### C.2 AUTO-TUNE PROCEDURE

During the auto-tuning procedure, we will tune the hyperparameters `sigma` and `n_iter_fwd`. For the remaining ones, we set `omega` and `n_iter_bwd` to their respective default values 1.7 and 25. For simplicity, we set `n_iter_test` to be equal to `n_iter_fwd`.

For the auto-tuning, we consider a subset of the validation set consisting of 150 contexts x out of the 1024 used in the benchmark problems of Section 3.1. Then, we generate 150 corresponding points-to-be-projected from a standard normal distribution $\mathcal{N}(0, I)$, which will serve as a proxy for the infeasible output of the backbone NN. First, to tune `sigma` we generate 100 logarithmically-spaced number between $10^{-3}$ and 5.05. Then, we compute the projection for each `sigma` on the 150 context–infeasible point pairs, by running 100 iterations of the projection layer. We evaluate the quality of each `sigma` by computing the resulting maximum constraint violation and average relative suboptimality of the projection, i.e., $\|z_K - y_{\mathrm{raw}}\| / \|\Pi_{\mathcal{C}(\mathbf{x})}(y_{\mathrm{raw}}) - y_{\mathrm{raw}}\|$. We consider any `sigma` for which both metrics are below certain thresholds to be candidate hyperparameters. We choose among the candidate hyperparameters the one with the minimum constraint violation.

We use the same procedure to choose `n_iter_fwd` from the set $\{50, 100, \ldots, 350, 400\}$, where the projection is evaluated using the previously determined `sigma`.

### C.3 ABLATIONS.

We evaluate the effectiveness of the matrix equilibration and auto-tuning by comparing the performance of our method with and without the different components on the large non-convex problem of Section 3.1. We test 3 different configurations: default hyperparameters and no equilibration; auto-tuned hyperparameters and no equilibration; auto-tuned hyperparameters and equilibration. We respectively refer to these configurations as `Default`, `Auto` and `Πnet`. On this problem setting, performing auto-tuning takes roughly 2 minutes, while the equilibration takes less than 1 second. The (tuned) hyperparameters for each configuration are as follows:

$$\mathtt{Default} \leftarrow \mathtt{sigma} = 1.0, \ \mathtt{n\_iter\_fwd} = 100$$
$$\mathtt{Auto} \leftarrow \mathtt{sigma} = 3.28, \ \mathtt{n\_iter\_fwd} = 350$$
$$\mathtt{\Pi net} \leftarrow \mathtt{sigma} = 0.161, \ \mathtt{n\_iter\_fwd} = 50$$

and the remaining ones are chosen as discussed previously. We note that the large difference in the value of `sigma` between `Auto` and `Πnet` is due to the equilibration that changes the problem scaling.

We report in Figure 16 the `RS` and `CV` by all 3 configurations, training curves are shown in Figure 17 for 50 training epochs and for a single seed, and inference times are given in Table 6. From Figure 16 we can readily deduce the substantial improvement in terms of `CV` that both auto-tuning and equilibration offers. Additionally, Figure 17 and Table 6 highlight the clear benefits of our preprocessing steps in terms of both training time and inference time. These benefits are due primarily to the fact that a well-tuned and equilibrated projection layer requires significantly fewer iterations to achieve a satisfactory `CV`, specifically 50 iterations instead of 100 and 350 for default and only auto-tuned, respectively.

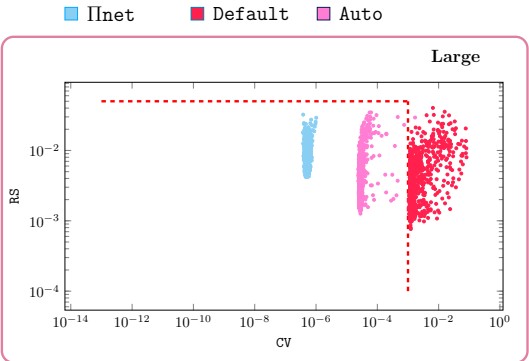

Figure 16: Visualization of the ablation results in Appendix C.3. Scatter plots of RS and CV for the methods on the large non-convex problems on the test set. The red dashed lines show the thresholds that we require to consider a candidate solution optimal.

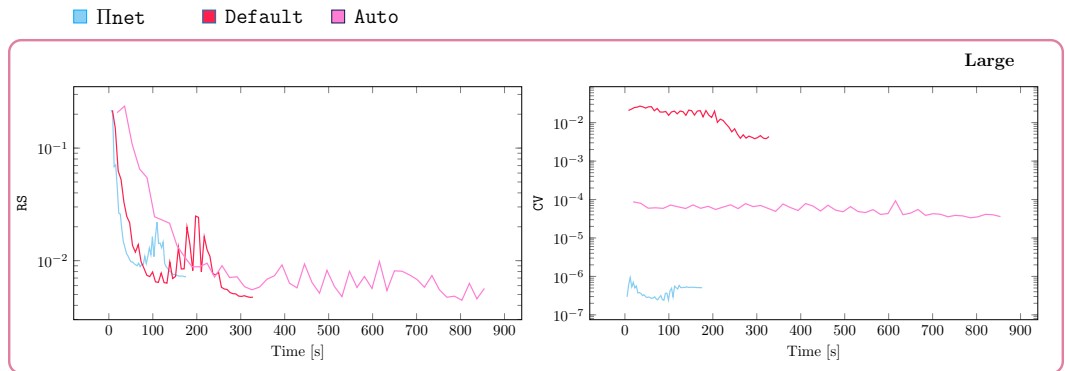

Figure 17: Comparison of the training times in terms of RS and CV for the different ablation configuration on the large non-convex problem setting. For simplicity, we report the learning curves only for a single seed.

| Method | Single inference [s] | | | | | Batch inference [s] | | | | |
|---|---|---|---|---|---|---|---|---|---|---|
| | median | LQ | UQ | min | max | median | LQ | UQ | min | max |
| **Non-convex large** | | | | | | | | | | |
| Default | 0.0080 | 0.0080 | 0.0081 | 0.0075 | 0.0090 | 0.5491 | 0.5490 | 0.5493 | 0.5487 | 0.5495 |
| Auto | 0.0141 | 0.0139 | 0.0143 | 0.0135 | 0.0150 | 1.8900 | 1.8899 | 1.8901 | 1.8896 | 1.8903 |
| Πnet | 0.0063 | 0.0063 | 0.0064 | 0.0061 | 0.0092 | 0.2804 | 0.2799 | 0.2807 | 0.2794 | 0.2912 |

Table 6: Inference time comparison for single-instance and batch-instance (1024 problems) settings across different ablation configurations, evaluated on the large non-convex problem. The table reports median runtime along with statistical descriptors: lower quartile (LQ, 25th percentile), upper quartile (UQ, 75th percentile), min and max of the runtime.

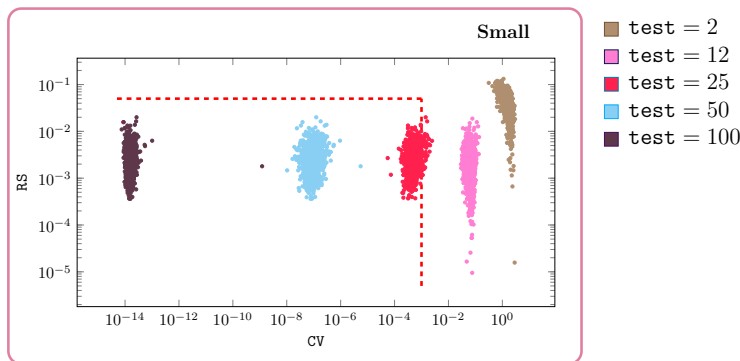

Figure 18: Scatter plot of `RS` and `CV` on the small non-convex test problems for a Πnet network trained with 50 iterations and evaluated with different numbers of iterations at test time. The red dashed lines indicate the thresholds used to consider a candidate solution optimal.

In our experiments, we have extensively tested the effects of using a different number of forward iterations during testing (`n_iter_test`) compared to the training (`n_iter_fwd`), with both more and fewer iterations at test time. Using more iterations works well and stably, and is meaningful to achieve reduced constraint violation. Using less iterations needs more care. Slightly less iterations is often possible and can improve inference time. Significantly less may cause issues since the iterates might not be close to a fixed-point anymore. Of course, "slightly" and "significantly" here depend on the problem at hand. For this, in Figure 18 we report the `RS` and `CV` obtained by training Πnet with `n_iter_fwd = 50`, and then deploying with different number of iterations during testing. Perhaps interestingly, note that the `RS` is unaffected by the number of iterations during testing, whereas the `CV` gradually decreases. This is true as long as we have enough iterations: as the number of iterations during test becomes too small (e.g., 2) also the `RS` is affected: The output is not close to the projection anymore.

Finally, in Figures 19 and 20 we report an ablation study on the parameters $\sigma$ and $\omega$. These results, together with the ones on the number of forward and backward iterations (cf. Figure 18 and Appendices D.3 and F), substantiate the claim of little sensitivity of Πnet to hyperparameter tuning. In particular, Figures 19 and 20 show that different values of the parameters $\sigma$ and $\omega$ yield qualitatively similar behaviors. For $\omega$, we see clearly that different values effectively only change the convergence rates: With more iterations, all values of $\omega$ achieve sufficiently low `CV`.

### C.4 Why enforcing constraints during training?

Πnet enforces the constraints during training, which warrants a discussion: What changes if one trains an unconstrained network and enforces the constraints only during inference? We answer this question with illustrative examples.

**Some optimization problems may cause divergence of the network if trained unconstrained.** The first example shows that, without constraints, the training process may even diverge, as the optimization problems we are interested in may not even be meaningful in the absence of constraints. Consider the parametric (in x) optimization problem:

$$\underset{y \in \mathbb{R}}{\text{minimize}} \quad xy \quad \text{subject to} \quad 0 \le y \le 1.$$

The optimal solution is

$$y^\star(x) = \begin{cases} 0 & \text{if } x < 0 \\ 1 & \text{if } x > 0 \\ \text{any } y \in [0,1] & \text{otherwise.} \end{cases}$$

However, if we would train an unconstrained network to predict $\hat{y}(x)$, its values would result in $\hat{y}(x) \to +\infty$ for $x > 0$ and $\hat{y}(x) \to -\infty$ for $x < 0$.

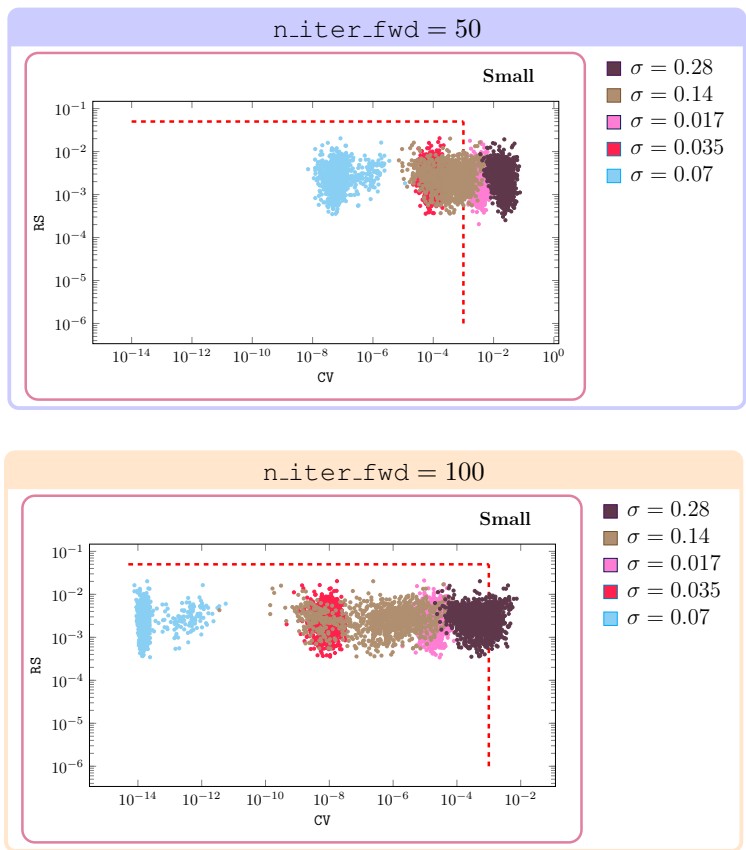

Figure 19: Scatter plot of RS and CV on the small non-convex test problems for a Πnet network trained with different values of $\sigma$ and (top) 50, (bottom) 100 forward iterations. The red dashed lines indicate the thresholds used to consider a candidate solution optimal.

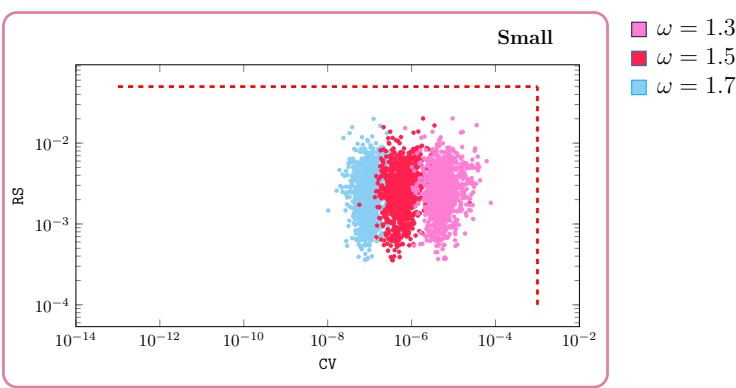

Figure 20: Scatter plot of RS and CV on the small non-convex test problems for a Πnet network trained with different values of $\omega$. The red dashed lines indicate the thresholds used to consider a candidate solution optimal.

It is easy to see how one may build higher-dimensional examples along the same lines of the presented one. Importantly, this example shows that, without constraints, the problem may be altered significantly to the point that it is not even possible to train an unconstrained network.

**The projection of the unconstrained optimizer is often suboptimal.** Our second example shows that applying the projection layer to the solution of the unconstrained problem does not yield, in general, the solution to the constrained problem.

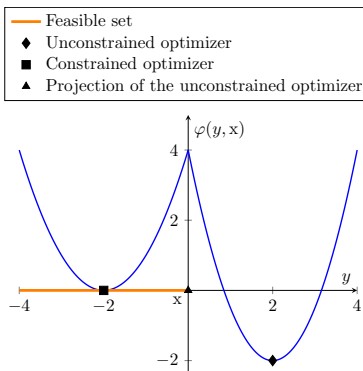

As a consequence, one can expect a strong decrease in performance when training the network unconstrained (i.e., to solve the unconstrained optimization problem) and only at inference adding a projection layer (i.e., projecting the solution of the unconstrained problem onto the feasible set). Consider the parametric (in $x$) optimization problem depicted on the right. Here, one may train an unconstrained network to optimize the objective function, finding the optimizer $\hat{y}(x) = x + 2$. However, projecting this onto the constraints $y \leq x$ would result in $y = x$, which is suboptimal compared to the constrained optimizer $y^\star(x) = x - 2$. Importantly, this simple example shows an instance in which training without constraints may downgrade the performances of the network.

**An empirical case-study.** In the third example, we train an unconstrained network on the small non-convex benchmark (using soft penalty terms) and add the projection layer only during inference on the test set (we refer to this as $\Pi$net-inf). The resulting average RS on the test set for $\Pi$net-inf is 0.02178, whereas for $\Pi$net it is 0.00216 using roughly the same computational budget (8 seconds of wall-clock time). The constraint violation is the same for both methods, since they use our projection layer during inference. Training with $\Pi$net (i.e., enforcing constraints during training) achieves one order of magnitude better RS. In fact, if we increase the computational budget, $\Pi$net continues to reduce the RS down to 0.0007 for 50 seconds of training. On the contrary, $\Pi$net-inf cannot reduce the RS further even with more training (or more hyperparameter tuning). Therefore, we observe in practice what our previous examples outlined: enforcing the constraints during training allows the network to anticipate the projection and further improve its predictions.

**Constraints as an inductive bias.** There is a subtle difference between constraining the architecture of a neural network (e.g., by ensuring that the neural network is a convex function for any values of the weights, as done by Amos et al. (2017)) and constraining the output to satisfy some properties without altering the "backbone" network architecture. In $\Pi$net, one can use the best existing architecture unaltered, but training it with the inductive bias that the output will be modified so to satisfy the constraints. For this, one may argue that enforcing the constraints during training improves the performance of the network at inference time, rather than downgrading it. We thus believe that an approach like $\Pi$net may, in fact, change the common perspective on the difficulty of training hard-constrained networks.

**Remark.** *It is perhaps worth clarifying the distinction between our work and the work of Amos et al. (2017):*

1. *Input convex neural networks learn a scalar-valued function, which is guaranteed to be a convex function of its input for any appropriate choices of the network weights (see Amos et al. (2017, Propositions 1, 2)). Then, during inference, a minimum of this function is computed. Recall that the set of minimizers of a convex function is a convex set, and in this sense $\Pi$net and input convex neural networks are similar.*

2. *On the contrary, $\Pi$net learns a vector-valued mapping which is guaranteed to lie on a convex set chosen a-priori for any choice of the weights.*

### C.5 $\Pi$NET AS AN IMPLICIT LAYER

The techniques used to implement $\Pi$net (operator-splitting and backpropagation via the implicit function theorem) are fundamental and widely adopted in the literature. In this sense, $\Pi$net is a special case of implicit layers (Agrawal et al., 2019; Butler & Kwon, 2023). However, our key ideas

are related to the type of problem we are solving (a projection) and how the structure of this problem can be exploited to derive an efficient formulation of the optimization algorithm. The improvement over the state of the art (as exemplified by our results in Section 3) is achieved by focusing on a problem setting that is sufficiently general yet rich in structure, and by adopting the right optimization techniques (e.g., which split to perform to deploy the Douglas-Rachford algorithm). The key ideas, in this sense, are:

1. Using a single additional layer instead of multiple ones.
2. Using a projection layer.
3. Adapting the Douglas-Rachford algorithm to best exploit the resulting problem structure.

# D DERIVATION DETAILS

In this section, we report the derivations omitted in the text.

## D.1 DERIVATION OF ALGORITHM 1.

To derive Algorithm 1 from (4), we will explicitly write the proximal operators in (4). For the $z$-update, we recall that $g = \mathcal{I}_{\mathcal{A}}$ and $\sigma > 0$, from which it immediately follows that $\text{prox}_{\sigma g} = \Pi_{\mathcal{A}}$. For the $t$-update, we recall that:

$$h\left(\begin{bmatrix} y \\ y_{\text{aux}} \end{bmatrix}\right) = \|y - y_{\text{raw}}\|^2 + \mathcal{I}_{\mathcal{K}}\left(\begin{bmatrix} y \\ y_{\text{aux}} \end{bmatrix}\right)$$

and using the definition of the proximal operator yields:

$$\text{prox}_{\sigma h}(2z_{k+1} - s_k) = \underset{y, y_{\text{aux}}}{\text{argmin}} \|y - y_{\text{raw}}\|^2 + \mathcal{I}_{\mathcal{K}}\left(\begin{bmatrix} y \\ y_{\text{aux}} \end{bmatrix}\right) + \frac{1}{2\sigma}\left\|\begin{bmatrix} y \\ y_{\text{aux}} \end{bmatrix} - 2z_{k+1} + s_k\right\|^2$$

$$= \begin{bmatrix} \underset{y \in \mathcal{K}_1}{\text{argmin}} \|y - y_{\text{raw}}\|^2 + \frac{1}{2\sigma}\|y - 2z_{k+1,1} + s_{k,1}\|^2 \\ \underset{y_{\text{aux}} \in \mathcal{K}_2}{\text{argmin}} \frac{1}{2\sigma}\|y_{\text{aux}} - 2z_{k+1,2} + s_{k,2}\|^2 \end{bmatrix}$$

$$= \begin{bmatrix} \Pi_{\mathcal{K}_1}\left(\frac{2z_{k+1,1} - s + 2\sigma y_{\text{raw}}}{1 + 2\sigma}\right) \\ \Pi_{\mathcal{K}_2}(2z_{k+1,2} - s_{k,2}) \end{bmatrix}.$$

## D.2 CONVERGENCE PROOF OF (4).

In this subsection, we show that (4) converges to a solution of (3). To do so, we first introduce the following mild assumption on the feasibility of the problem.

**Assumption 1.** $\mathcal{A} \cap \text{ri}\,\mathcal{K} \neq \emptyset$, where $\text{ri}$ is the relative interior.

This assumption corresponds to strict feasibility of (3). In fact, if $\mathcal{K}$ is a polyhedron, then we can relax Assumption 1 to $\mathcal{A} \cap \mathcal{K} \neq \emptyset$.

To obtain our convergence result, we simply note that iteration (4) is the Douglas-Rachford algorithm applied to problem (3) and, under Assumption 1, we invoke the Corollaries 27.6 and 28.3 in (Bauschke & Combettes, 2017) which yield the desired result.

## D.3 CONVERGENCE RATES OF (4).

The convergence rate of the Douglas-Rachford algorithm is a well-studied topic in the literature, which we report here for completeness. For the case of a convex QP, it has been shown that the Douglas-Rachford algorithm attains a linear convergence rate (Peña et al., 2021, Subsection 3.2), namely an $\varepsilon$-optimal solution is computed in $\mathcal{O}(\log(1/\varepsilon))$ iterations. This result applies to our algorithm in the case of polyhedral constraints, and indeed we have observed these rates in our numerical experiments (see also Figure 21). Other technical conditions for linear convergence are given in (Hong & Luo, 2017, Assumption A); their applicability to our setting depends on the specific constraint set $\mathcal{C}$. In the general setting, the Douglas-Rachford algorithm typically attains a $\mathcal{O}(1/k)$ convergence rate (where $k$ is the number of iterations), hence requiring $\mathcal{O}(1/\varepsilon)$ iterations (Davis & Yin, 2017).

We visualize these convergence rates in the plot in Figure 21, showing how for increasing iterations we get a rapidly decreasing CV. Specifically, for one of the benchmarks, we provide the CV for 100 instances in the test set for an increasing number of iterations. In particular, we observe that within very few iterations the constraints are satisfied up to high accuracy (e.g., $10^{-5}$ as in (Stellato et al., 2020)).

## D.4 DERIVATION OF THE BACKWARD PASS

Applying the implicit function theorem to $s_{\infty}(y_{\text{raw}}) = \Phi(s_{\infty}(y_{\text{raw}}), y_{\text{raw}})$ yields the linear equation:

$$\left(I - \left.\frac{\partial\Phi(s, y_{\text{raw}})}{\partial s}\right|_{s=s_{\infty}(y_{\text{raw}})}\right)\left.\frac{\partial s_{\infty}(y_{\text{raw}})}{\partial y_{\text{raw}}}\right|_{y_{\text{raw}}=f(x;\theta)} = \left.\frac{\partial\Phi(s, y_{\text{raw}})}{\partial y_{\text{raw}}}\right|_{y_{\text{raw}}=f(x;\theta)}, \quad (15)$$

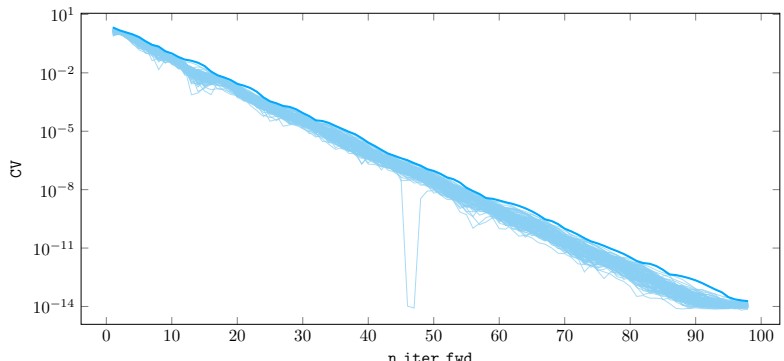

Figure 21: Constraint violation for 100 problem instances (light blue) in the test set of the small non-convex benchmark as the number of iterations in the forward pass `n_iter_fwd` (i.e., `n_iter_train` increases, as well as the maximum CV among all instances (dark blue).

whose unknown is the Jacobian matrix $\frac{\partial s_\infty(y_{\text{raw}})}{\partial y_{\text{raw}}}\big|_{y_{\text{raw}}=f(\text{x};\theta)}$.

Now, to compute the VJP

$$v^\top \frac{\partial s_\infty(y_{\text{raw}})}{\partial y_{\text{raw}}}\bigg|_{y_{\text{raw}}=f(\text{x};\theta)}$$

where $v \in \mathbb{R}^{n+d}$, we assume that the linear system (9) admits a solution, namely $\xi(y_{\text{raw}}, v)$. Next, we pre-multiply (15) by $\xi(y_{\text{raw}}, v)^\top$:

$$\xi(y_{\text{raw}}, v)^\top \left( I - \frac{\partial \Phi(s, y_{\text{raw}})}{\partial s}\bigg|_{s=s_\infty(y_{\text{raw}})} \right) \frac{\partial s_\infty(y_{\text{raw}})}{\partial y_{\text{raw}}}\bigg|_{y_{\text{raw}}=f(\text{x};\theta)} = \xi(y_{\text{raw}}, v)^\top \frac{\partial \Phi(s, y_{\text{raw}})}{\partial y_{\text{raw}}}\bigg|_{y_{\text{raw}}=f(\text{x};\theta)}$$

$$\implies v^\top \frac{\partial s_\infty(y_{\text{raw}})}{\partial y_{\text{raw}}}\bigg|_{y_{\text{raw}}=f(\text{x};\theta)} = \xi(y_{\text{raw}}, v)^\top \frac{\partial \Phi(s, y_{\text{raw}})}{\partial y_{\text{raw}}}\bigg|_{y_{\text{raw}}=f(\text{x};\theta)}$$

where the implication follows by the definition of $\xi(y_{\text{raw}}, v)$. This gives rise to our backward pass outlined in (8) and (9).

### D.5   COMPUTATIONAL COMPLEXITY OF $\Pi$NET.

Here we derive the computational complexity of $\Pi$net for both the forward and backward pass.

**Forward.**   The complexity of the projection operations in Algorithm 1 are determined by[1] the sets $\mathcal{A}(\text{x})$, $\mathcal{K}_1(\text{x})$ and $\mathcal{K}_2(\text{x})$, by the cost of instantiating them given the context x, which may depend on the dimensionality $p$ of x, and by the dimensions of the input $n$, $d$ and $n - d$. We denote the complexity of these operations as $g_{\text{x}}(n, p), g_\mathcal{A}(n), g_{\mathcal{K}_1}(d)$ and $g_{\mathcal{K}_2}(n - d)$. The overall complexity is:

$$\mathcal{O}\left(g_{\text{x}}(n, p) + K(g_\mathcal{A}(n) + g_{\mathcal{K}_1}(d) + g_{\mathcal{K}_2}(n - d) + n)\right),$$

where $K = $ n_iter_fwd is the number of forward iterations, and the term $n$ is due to the sum[2] in the governing update (4c).

Next, we analyze the complexity of $g_{\text{x}}(n, p), g_\mathcal{A}(n), g_{\mathcal{K}_1}(d)$ and $g_{\mathcal{K}_2}(n - d)$:

- $g_{\text{x}}(n, p)$ is problem specific. In the examples presented in this work, the complexity is linear in $p$ and amounts to stacking the vectors describing the problem instance (e.g., the initial and final position of the vehicles in Section 3.2) into the equality constraints. In general, one may expect $g_{\text{x}}(n, p)$ to be dominated by the other contributions. If the matrix $A$ is context dependent, an additional cost is the calculation of the pseudo-inverse, which we use to compute the projection

---

[1]The complexity also depends on the batch size, but the operations can be fully parallelized over a batch and, thus, we focus on the per-element complexity.

[2]On a GPU, this entry-wise vector sum can be parallelized, but here we focus on the number of operations.

onto the hyperplane: $g_{\mathrm{x}}(n, p) = n^3$; see also the next item. When the matrix $A$ is independent of the context, the pseudo-inverse can be computed only once at the instantiation of the method, and its computational complexity does not affect the forward pass.

- $g_{\mathcal{A}}(n)$ is the cost of the projection on the hyperplane identified by a matrix $A \in \mathbb{R}^{(d+d') \times n}$ and a vector $b \in \mathbb{R}^{d+d'}$. The parameter $d'$ is problem-dependent, but usually proportional to the number of inequality constraints; see Appendix E. For this reason, we carry out the complexity analysis for $A \in \mathbb{R}^{n \times n}$. For the projection, we need the pseudo-inverse of $A$, which we compute once for all iterations. Thus, we account for this complexity in $g_{\mathrm{x}}(n, p)$. Then, the complexity of computing the projection is dominated by matrix-vector multiplications: $g_{\mathcal{A}}(n) = n^2$.

- $g_{\mathcal{K}_1}(d)$ and $g_{\mathcal{K}_2}(n - d)$ are the cost of the projection onto a box after a linear combination of the input. Thus, $g_{\mathcal{K}_1}(d) + g_{\mathcal{K}_2}(n - d) = n$.

Overall, the complexity of the forward pass is $\mathcal{O}(n^2)$ for context-independent $A$ and $\mathcal{O}(n^3)$ for context-dependent $A$. On a GPU and for context-independent $A$ matrices, the effective compute time is possibly linear in $n$.

**Backward.** For the backward pass, we need to:

- Compute the $v$ for the linear system in (9), for which we need only the first VJP in (7). The complexity is $\mathcal{O}(d(n + d)) = \mathcal{O}(n^2)$.

- Solve the linear system in (9) using `bicgstab` (van der Vorst, 1992). The complexity of this operation is $\mathcal{O}(K'(n + d)d) = \mathcal{O}(K'n^2)$, where $K' = $ `n_iter_backward` is the number of iterations used for `bicgstab`.

- We compute the VJP in (8). By direct inspection of the computational graph of (4a)-(4c), complexity of this step is $\mathcal{O}(d(d + n)) = \mathcal{O}(n^2)$.

Overall, the backpropagation through our projection layer has complexity $\mathcal{O}(n^2)$, with the constant dominated by the number of iterations used for `bicgstab`.

# E  MORE EXAMPLES OF CONSTRAINT SETS

In this section, we describe how several classes of constraints that can be described within our framework admit an efficient projection. We stress that the decomposition $\mathcal{C} = \Pi_d(\mathcal{A} \cap \mathcal{K})$ is not an assumption. One can always decompose a convex set in this way, e.g., by considering the trivial decomposition $\mathcal{A} = \mathcal{C}$ and $\mathcal{K} = \mathbb{R}^d$. Instead, determining $\mathcal{A}$ and $\mathcal{K}$ is a design choice which we leverage to make the projections $\Pi_\mathcal{A}$ and $\Pi_\mathcal{K}$ computationally efficient. We note two important points regarding this design choice:

- The only assumption is that $\Pi_\mathcal{A}$ and $\Pi_\mathcal{K}$ and their VJP are computable. Being computationally efficient is an added benefit of our decomposition, but is not necessary.

- Below, we show that many practically-relevant constraints $\mathcal{C}$ can be decomposed in a computationally-efficient manner: polyhedra, second-order cones, sparsity constraints, simplices, and the intersections and Cartesian products all admit efficient decompositions. In fact, this list is not exhaustive; see, e.g., (Condat, 2016; Boyd & Vandenberghe, 2004).

We believe the implementation and adoption of these constraints in practical applications represents an important direction for future work.

**Polytopic sets**  are often employed to enforce constraints in robotics (Augugliaro et al., 2012), numerical solutions to PDE (Raissi et al., 2019), and non-convex relaxations for trajectory planning (Malyuta et al., 2022), among others. They are expressed as

$$\{y \in \mathbb{R}^d \mid Ey = q, l \leq Cy \leq u\},$$

for some $E, q, l, C, u$ of appropriate dimensions. We introduce the auxiliary variable $y_{\text{aux}} = Cy \in \mathbb{R}^{n_{\text{ineq}}}$ with dimension $n - d = n_{\text{ineq}}$. Then, we respectively define $\mathcal{A}, \mathcal{K} \subseteq \mathbb{R}^n$ as the following affine subspace and box

$$\mathcal{A} = \left\{ \begin{bmatrix} y \\ y_{\text{aux}} \end{bmatrix} \,\middle|\, \underbrace{\begin{bmatrix} E & 0 \\ C & -I \end{bmatrix}}_{=A} \begin{bmatrix} y \\ y_{\text{aux}} \end{bmatrix} = \underbrace{\begin{bmatrix} q \\ 0 \end{bmatrix}}_{=b} \right\}, \quad \mathcal{K} = \left\{ \begin{bmatrix} y \\ y_{\text{aux}} \end{bmatrix} \,\middle|\, y \in \mathbb{R}^d, \, l \leq y_{\text{aux}} \leq u \right\}.$$

Importantly, $\mathcal{C} = \Pi_d(\mathcal{A} \cap \mathcal{K})$ and both $\Pi_\mathcal{A}$ and $\Pi_\mathcal{K}$ can be evaluated in closed form (Bauschke & Combettes, 2017, Chapter 29).

**Second-order cones**  are employed in portfolio optimization (Brodie et al., 2009), robust control (Chen et al., 2018), and support vector machines (Maldonado & López, 2014), among others. They involve constraints of the form:

$$\{y \in \mathbb{R}^d \mid \|Cy + c\|_2 \leq f^\top y + e\}.$$

Introducing the auxiliary variables $y_{\text{aux},1} = Cy + c \in \mathbb{R}^{n_c}$ and $y_{\text{aux},2} = f^\top y + e \in \mathbb{R}$, of dimension $n - d = n_c + 1$, we obtain the representation:

$$\mathcal{A} = \left\{ \begin{bmatrix} y \\ y_{\text{aux},1} \\ y_{\text{aux},2} \end{bmatrix} \,\middle|\, \underbrace{\begin{bmatrix} C & -I & 0 \\ f^\top & 0 & -1 \end{bmatrix}}_{=A} \begin{bmatrix} y \\ y_{\text{aux},1} \\ y_{\text{aux},2} \end{bmatrix} = \underbrace{\begin{bmatrix} -c \\ -e \end{bmatrix}}_{=b} \right\}, \quad \mathcal{K} = \left\{ \begin{bmatrix} y \\ y_{\text{aux},1} \\ y_{\text{aux},2} \end{bmatrix} \,\middle|\, \|y_{\text{aux},1}\|_2 \leq y_{\text{aux},2} \right\}$$

for quantities of the appropriate dimensions. Again, $\Pi_\mathcal{A}$ and $\Pi_\mathcal{K}$ admit a closed form solution (Boyd & Vandenberghe, 2004).

**Sparsity constraints**  encourage solutions with fewer active variables, such as the number of open positions in portfolio optimization (Brodie et al., 2009), or the "mass splitting" in optimal transport (Dantzig, 2002). Explicitly, they read

$$\{y \in \mathbb{R}^d \mid \|Cy + c\|_1 \leq f^\top y + e\}$$

Analogously to second-order cones constraints, employing a slack variable they can be reduced to a projection onto the $l_1$ ball, which can be done efficiently (Condat, 2016).

**Cartesian products of simplices** are the standard constraints in optimal transport (Dantzig, 2002) formulations, since they encode the set of admissible coupling measures. While these constraints can also be expressed as generic polytopic sets, an alternative representation using our proposed form can potentially yield more efficient solutions. For instance, for some $v_1, \ldots, v_n, w_1, \ldots, w_m \geq 0$, consider the following constraint set

$$
\mathcal{C} = \left\{ y \in \mathbb{R}^{d_1 \times d_2} \,|\, y_{ij} \geq 0, \sum_{i=1}^{n} y_{ij} = w_j, \sum_{j=1}^{m} y_{ij} = v_i \right\}
$$

$$
= \left\{ y \in \mathbb{R}^{d_1 \times d_2} \,|\, \sum_{i=1}^{n} y_{ij} = w_j \right\} \cap \left\{ y \in \mathbb{R}^{d_1 \times d_2} \,|\, y_{ij} \geq 0, \sum_{j=1}^{m} y_{ij} = v_i \right\}.
$$

$$
= \mathcal{A} \cap \mathcal{B}
$$

Notice, that $\mathcal{C} = \Pi_d(\mathcal{A} \cap \mathcal{K})$ as required by our constraint decomposition, and $\Pi_d(\cdot)$ is redundant since $\mathcal{A}$ and $\mathcal{K}$ are of the same dimension as $\mathcal{C}$. The projections $\Pi_\mathcal{A}$ and $\Pi_\mathcal{K}$ can be evaluated efficiently since $\mathcal{A}$ is a hyperplane and $\mathcal{K}$ is a cartesian product of simplices (Condat, 2016).

**The intersection of previous sets** can readily be expressed in our representation. For instance, the intersection of a polytope and second-order cone can be expressed as the following set:

$$
\{ y \in \mathbb{R}^d \,|\, Ey = q, \, l \leq Cy \leq u, \, \|Fy + c\|_2 \leq f^\top y + e \}.
$$

We introduce the auxiliary variables $y_{\text{aux}, 1} = Cy \in \mathbb{R}^{n_{\text{ineq}}}$, $y_{\text{aux}, 2} = Fy \in \mathbb{R}^{n_c}$, $y_{\text{aux}, 3} = f^\top y \in \mathbb{R}$ and obtain the representation:

$$
\mathcal{A} = \left\{ \begin{bmatrix} y \\ y_{\text{aux}, 1} \\ y_{\text{aux}, 2} \\ y_{\text{aux}, 3} \end{bmatrix} \,\middle|\, \begin{bmatrix} E & 0 & 0 & 0 \\ C & -I & 0 & 0 \\ F & 0 & -I & 0 \\ f^\top & 0 & 0 & -1 \end{bmatrix} \begin{bmatrix} y \\ y_{\text{aux}, 1} \\ y_{\text{aux}, 2} \\ y_{\text{aux}, 3} \end{bmatrix} = \begin{bmatrix} q \\ 0 \\ -c \\ -e \end{bmatrix} \right\},
$$

$$
\mathcal{K} = \mathbb{R}^d \times \{ y_{\text{aux}, 1} \in \mathbb{R}^{n_{\text{ineq}}} \,|\, l \leq y_{\text{aux}, 1} \leq u \} \times \left\{ \begin{bmatrix} y_{\text{aux}, 2} \\ y_{\text{aux}, 3} \end{bmatrix} \in \mathbb{R}^{n_c + 1} \,|\, \|y_{\text{aux}, 2}\| \leq y_{\text{aux}, 3} \right\}.
$$

As before, the projections $\Pi_\mathcal{A}$ and $\Pi_\mathcal{K}$ admit closed-form expressions.

# F ON THE DIFFERENTIABILITY OF THE PROJECTION LAYER AND THE APPLICABILITY OF THE IMPLICIT FUNCTION THEOREM

In this section, we discuss the theoretical aspects of the differentiability of the proposed projection layer, as well as the applicability of the implicit function theorem.

## F.1 ALMOST EVERYWHERE DIFFERENTIABILITY

To start, we note that since the layers of the backbone network will be at most almost everywhere differentiable (e.g., for `ReLU` activations), all one may be interested in is to have almost everywhere differentiability of the projection layer. In fact, our experiments show that in practice this is sufficient for markedly surpassing the state of the art. On Euclidean spaces (more generally, on Hilbert spaces) the projection operator is globally 1-Lipschitz (see, e.g., (Bauschke & Combettes, 2017, Proposition 4.16), and recall that firmly non-expansive implies 1-Lipschitz). Then, with Rademacher's theorem (Simon et al., 1984, Theorem 1.4) we conclude that the projection layer is almost everywhere differentiable (in the Lebesgue measure sense).

## F.2 APPLICABILITY OF THE IMPLICIT FUNCTION THEOREM

Providing a full theoretical analysis is beyond the scope of the paper, and we believe it would distract the reader from the main message of the paper. Nonetheless, we (i) believe this to be a very interesting mathematical quest for future work, and (ii) that the outline presented here can give a sense to the reader of why the proposed projection layer is theoretically sound.

The implicit function theorem is applicable under local differentiability conditions. The way that related works handle non-smooth points is to either (i) show that if the linear system (9) is non-singular, then multiple solutions exist and they belong to the subdifferential (see, e.g., (Amos & Kolter, 2017, Appendix C.1)) or (ii) they resort to heuristics, e.g., by solving a least-squares approximation rather than the linear system itself (see, e.g., (Agrawal et al., 2019, Section 4.3)). We similarly run finitely many `bicgstab` iterations, which is very computationally attractive as it avoids run-time checks and, in our numerical experience, is quite stable; see Appendix F.3.

Perhaps surprisingly, one can define an implicit function theorem even on non-smooth points by employing *conservative gradients*, a recent generalization of the subdifferential (Bolte & Pauwels, 2021). This formulation is valid for non-smooth locally Lipschitz mappings, to which our projection layer belongs. We refer to the seminal works of Bolte & Pauwels (2021); Bolte et al. (2021) for the precise statement of the non-smooth variant of the theorem. In this framework, the implicit function theorem formula and the chain rule used in Section 2.1.2 hold under very mild conditions. Roughly speaking, this would correspond to our operator $\Phi(x, y)$ being semialgebraic and contractive with respect to $y$ (Bolte et al., 2024). With this, we could conclude that our derivation in Section 2.1.2 holds everywhere.

## F.3 EFFECTIVENESS OF THE BACKWARD PASS IN APPROXIMATING THE GRADIENT

In this section, we empirically assess how well the proposed backward pass approximates the true gradient induced by the projection operator. Concretely, we compare the VJP obtained from the implicit function theorem–based backward pass with finite-difference estimates on a representative problem instance.

**Experimental setup** We fix two problem instances in the non-convex small benchmark (see Section 3.1) by fixing the right-hand side of the equality constraints. We then generate a set of 100 points-to-be-projected $x_i{}_{i=1}^{100}$ by sampling each coordinate from a normal distribution with standard deviation 10. For each $x_i$, we also sample 100 normally distributed vectors $v_i$ for comparing the VJP. As a reference "ground truth", we compute the VJP using central finite differences with step size $\varepsilon = 10^{-6}$, which yields an approximation with $\mathcal{O}(\varepsilon^2)$ truncation error. We then compute the VJP using our backward pass for varying numbers of backward iterations `n_iter_bwd`. For each configuration, we report (i) the $\ell_2$-norm of the difference and (ii) the cosine similarity, thereby quantifying both the absolute and directional accuracy of the proposed backward pass. For both methods, we use `n_iter_fwd` = 200 forward iterations for computing the projection accurately.

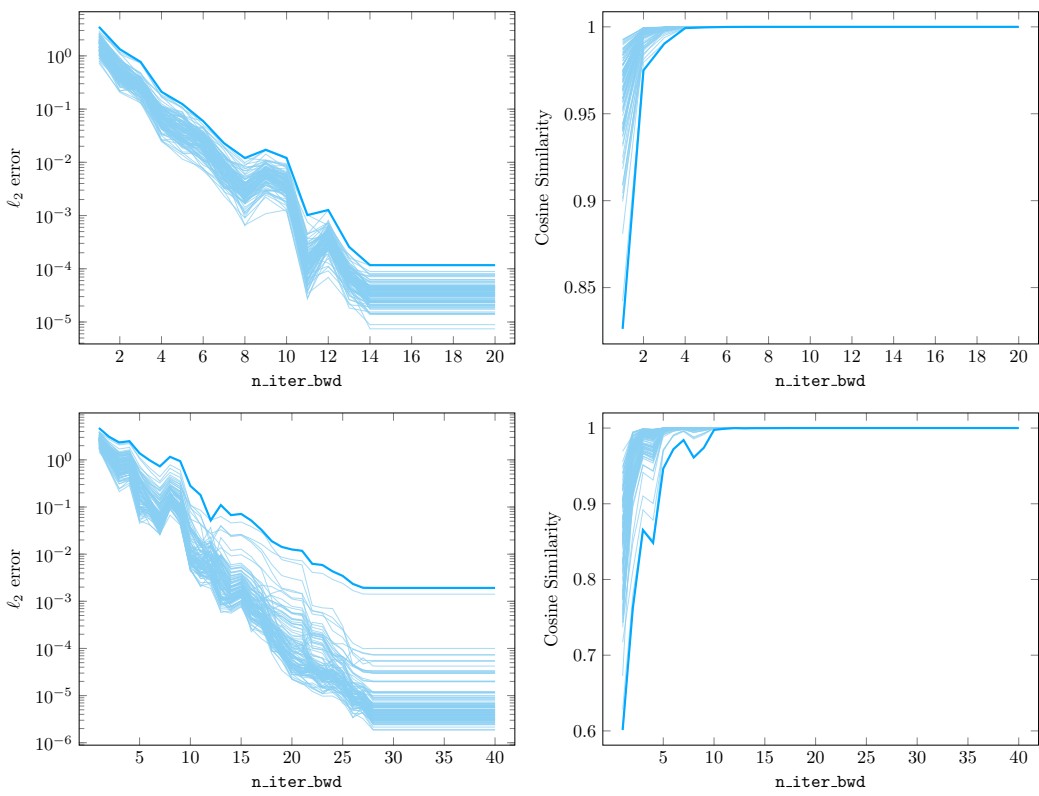

Figure 22: VJP estimation error (measured via the $\ell_2$-norm of the difference and the cosine similarity) for 100 different vectors $v_i$ instances (light blue) for two different instances of the small non-convex benchmark as the number of iterations in the backward pass `n_iter_bwd` increases, as well as the maximum $\ell_2$-error and the minimum cosine similarity among all instances (dark blue).

**Results**    As shown in Figure 22, the discrepancy between the VJP computed with our backward pass and the finite-difference reference decreases rapidly as the number of backward iterations increases. These results indicate that the proposed backward pass yields a highly accurate approximation of the true gradient even with a relatively small number of iterations.

