# OpenReview forum: "Pinet: Optimizing hard-constrained neural networks with orthogonal projection layers"
_ICLR.cc/2026/Conference — ICLR 2026 Oral_

### Official Review · Reviewer_1VJT · 2025-10-29

**Soundness:** 3
**Presentation:** 2
**Contribution:** 2
**Rating:** 6
**Confidence:** 2

**Summary:**

This paper introduces \PiNet, a neural network architecture that guarantees hard constraint satisfaction by projecting the network’s output onto a convex feasible set using operator splitting (Douglas–Rachford iterations) in the forward pass and applying the implicit function theorem (IFT) for efficient differentiation in the backward pass. ΠNet acts as a feasible-by-design implicit layer, enabling neural networks to directly solve parametric constrained optimization problems. The method is further applied to multi-vehicle motion planning, showcasing constraint satisfaction, parallelizability, and flexibility to optimize arbitrary differentiable objectives. A JAX implementation is provided for GPU-ready deployment.

**Strengths:**

ΠNet introduces a principled mechanism for embedding orthogonal projection layers into neural networks, rigorously ensuring constraint satisfaction during both training and inference.

The layer can be seamlessly attached to any backbone network, transforming unconstrained predictors into feasible solution generators for convex programs, PDEs, and motion-planning tasks.

**Weaknesses:**

ΠNet cannot directly handle non-convex or mixed-integer constraints; while the authors mention sequential convexification as future work, this remains a strong limitation for broader adoption.

While convergence of the projection layer is discussed, formal proofs of differentiability stability and error propagation across finite iterations are missing.

The multi-vehicle experiment is insightful but limited. No comparison is made with structured planners (e.g., QP-based MPC, or differentiable MPC variants).

**Questions:**

How does ΠNet perform relative to OptNet or Theseus, which also differentiate through optimization problems?

How many Douglas–Rachford iterations are typically needed for practical convergence? Is there a risk of slow progress or oscillation for poorly conditioned problems?

In multi-vehicle planning, each vehicle’s constraints are decoupled. How would ΠNet behave when inter-agent constraints (e.g., collision avoidance) are included?

The paper claims strong insensitivity to tuning—could you provide more quantitative evidence or stress tests across random seeds and scaling factors?

---

> ### Author Response · Authors · 2025-11-21
> **Response from Authors to Reviewer 1VJT - Weaknesses**
>
> We thank the reviewer for the questions. We also kindly ask the reviewer to point us why they consider our exposition (presentation) "fair". We have put substantial amount of work to clarify the exposition and we believe it is more in line with the scores from other reviewers, who considered the paper well written, clear, and with consistent notation. We would deeply appreciate suggestions to improve that score.
>
> **Weaknesses**
>
> - We agree that convexity of the constraints is a limitation of our work. However, it is worth noticing that convex optimization has revolutionized numerous fields in practice, so we respectfully disagree that this limitation poses a major obstacle to broader adoption. With respect to sequential convexification, the page limit and the necessity for in-depth analysis already for the convex case make it so that including it in the same work is not a feasible decision (assuming no compromises in quality and clarity). Sequential convexification has several building blocks that need to be worked out for having a framework that actually works. The major such block is a solver for the convex subproblems, which is the focus of this work. Having an unreliable convex solver would hamper any sequential convexification effort. As such, we do not see our work as limited, but rather as an extensive and precise solution to this first problem, on top of which sequential convexification can be developed. In the revised manuscript, we have extended the discussion upon the generality of our method (see rev. o3mf).
> - We are not sure we have interpreted this weakness correctly, so we kindly ask the reviewer to clarify it so that we can properly address it if we have not done so. Assuming the differentiability refers to the differentiability of the projection layer, we have now added an entire appendix devoted to this aspect (Appendix F). We kindly ask the reviewer to also refer to the answer given to rev. yrzg. The same appendix also includes analysis with respect to what happens with a finite number of backward iterations, and in Appendix D.2 and D.3 we provide convergence proofs, rates, and empirical evidence for the forward pass. We are not sure what "stability" refers to, so we kindly ask the reviewer to clarify.
> - We thank the reviewer for the chance of clarifying that QP-based methods are not applicable to our setup. The main objective of the multi-agent planning method is to showcase problems that cannot be solved via, e.g., QP but can be solved with our method.

---

> ### Author Response · Authors · 2025-11-21
> **Response from Authors to Reviewer 1VJT - Questions**
>
> **Questions**
>
> - Regarding OptNet, we have not made a comparison with this work. Instead, we compare with the JAXopt implementation of OSQP, which is also a differentiable QP solver (same as OptNet). JAXopt is more recent than OptNet and implemented in JAX rather than PyTorch, so we believe that it is an equal if not stronger baseline. Finally, our results show that $\Pi$net significantly outperforms JAXopt. Regarding Theseus, our works are not comparable since Theseus is a differentiable non-linear least squares solver. No constraints are considered in that work, as opposed to $\Pi$net which is focused on enforcing constraints. In fact, the authors of Theseus mention this as one of the limitations of their work in the paper’s Conclusion.
> - The exact number of iterations may vary between problems. In the various problems we consider, the number of iterations is between 50 and 1000, depending on the problem and desired accuracy. As for all first-order methods, indeed the DR algorithm may experience slow progress (or oscillations) for poorly conditioned problems. We have implemented two practical measures to counter this effect:
>     - Matrix equilibration, to improve the condition of the problem.
>     - Auto-tuning, to automatically choose performant hyperparameters for the projection layer (before training the network).
>
>     We verify their efficacy in Appendix C.3. We show that they significantly reduce the required number of iterations, they reduce constraint violations, and they speed-up training and inference. Please refer to Figures 14 and 15 for more details. Convergence proofs, rates, and empirical effects of the number of forward iterations are provided in Appendices D.2 and D.3. We study the effects of a finite number of backward iterations in the newly added Appendix F.
>
> - We thank the reviewer for the question. We can handle coupled constraints, by solving one larger projection problem. This is still possible but might reduce efficiency. Collision constraints are non-convex, so they fall in the category of problems that we should solve with sequential convexification. Other coupled, convex constraints can be handled with formulations similar to the ones in the benchmarks. Depending on the time and all the other benchmarks we are performing during this rebuttal, we may add an instance of such coupled constraints, if this would make the reviewer consider increasing the score of the paper.
> - We thank the reviewer for this question, since it allows us to elaborate more on this aspect. To start, we would like to clarify this sentence from the reviewer: "The paper claims strong insensitivity to tuning". We do not claim that, but rather we claim less sensitivity to hyperparameter tuning compared to existing methods. This of course is only related to the way we ensure a backbone network satisfies the constraints (a vanilla MLP as  backbone network will still be sensitive to hyperparameters such as learning rate). We perform several ablations additionally to the ones already available for showcasing how the hyperparameters affect our results. Specifically, we provide ablations on:
>
>      - the number of forward iterations (how CV rapidly decreases with increasing forward iterations)
>      - backward iterations (how our estimated gradient rapidly approaches the one obtained by finite differences)
>      - sigma, with 50 and 100 iterations
>      - omega
>
> Plots available (anonymously) at: https://imgur.com/a/oEUSe9z
> We integrated these ablations in the revised manuscript (changes highlighted in red).

---

### Official Review · Reviewer_yrzg · 2025-10-29

**Soundness:** 3
**Presentation:** 3
**Contribution:** 2
**Rating:** 6
**Confidence:** 3

**Summary:**

This paper introduces Πnet, a neural network architecture that generates provably feasible solutions for parametric constrained optimization problems. The core innovation is a projection layer that maps a backbone network's raw output onto a convex feasible set using the Douglas-Rachford algorithm, guaranteeing hard constraint satisfaction by design. A key advantage is its efficient training via the implicit function theorem for backpropagation, avoiding the computational cost of unrolling the optimization.

**Strengths:**

1)  А mathematically sound appendable layer with a robust Forward pass formulation and locally correct and heuristically consistent Backward pass.
2)  Empirical results are meaningful and appear to be reproducible. The ablation study is sufficient.
3)  Even with approximate or subgradient differentiation, the forward pass is a valid projected inference scheme. Douglas–Rachford ensures convergence under mild conditions.
4)  The backward pass gives a practical, stable gradient proxy. Empirically, autodiff through a few DR steps often works and trains faster than unrolled solvers.
5)  The extra projection layer adds no extra parameters, and only a moderate computational overhead—roughly 2–3x vs an unconstrained MLP, far less than unrolled solvers or cvxpylayers.

**Weaknesses:**

My main concern is the differentiability during training passes, which may make the Pinet functional only locally.  Please review the mathematical assumptions required for the method to work:
-- C(x) convex, closed, and with a fixed active constraint set around each training point (no switching during differentiation);
-- A(x) full row rank;
-- Φ(s,y) differentiable in a neighborhood of the fixed point (piecewise but locally nonsingular (local contraction);
It is my understanding that only then the implicit-function theorem can apply locally, and the gradient formula (Eq. 8) would become correct within that region.
Likewise, in appendix, no derivations of the explicit Jacobians are provided. For the IFT to apply, the map $\Phi(s, y_{\text{raw}})$ must be differentiable. The authors never show $\partial_s \Pi_A$ or $\partial_s \Pi_K$.

The authors also start from $s_\infty = \Phi(s_\infty, y_{\text{raw}})$ and derive $$(I - \partial_s \Phi)^\top \xi = v, \quad v^\top \partial_{y_{\text{raw}}} \Phi$$, which is algebraically correct; however, the solution might diverge if $I - \partial_s \Phi$ is singular (non-invertible), and the numerical solution via Bi-CGSTAB may be wrong. Also, if $\Phi$ is built from non-smooth proximal operators it is obviously not $C^1$. Please clarify whether the solution is stable and unique.

Lines 161–163: The projection map is non-differentiable at points where the active constraint set changes. The paper later differentiates this function as if it were $C^1$. So, one needs to clarify that it is Lipschitz but not differentiable everywhere; and the gradients are defined only almost everywhere or as subgradients.

Lines 173–179: Projection commutativity is assumed without proof. Derive or restrict A,K so that projection decomposition preserves convex geometry; otherwise (2) does not yield the same minimizer as (1).

Eqs. (3a–3c): replace “strict feasibility” with “non-empty intersection and standard monotonicity assumptions.”, because DR requires only non-empty intersection of closed convex sets. Also, include the σ scaling in prox definitions (currently inconsistent with (3a, 3b)).

Lines 218–228: IFT application will be valid only where projection is differentiable (interior or non-degenerate active set). This assumption is implicit and needs to be discussed.

Recommendations:

•  Replace global claims (“guaranteed differentiability”) with local or empirical ones. Framing results as empirical efficiency rather than mathematical guarantee is advised.
•  Include a small proof or citation showing IFT validity for smooth convex sets with full-rank equality constraints.

**Questions:**

Given the significant challenge of enforcing the strict constraints in large-scale or parametric Optimal Transport (OT) problems, how does the Πnet architecture with its hard-constrained compare to state-of-the-art OT solvers like the Sinkhorn algorithm and Adaptive Primal-Dual Accelerated Gradient Descent (APDAGD)?

---

> ### Author Response · Authors · 2025-11-21
> **Response of Authors to Reviewer yrzg**
>
> We thank the reviewer for their comments. We believe to have thoroughly addressed all the comments (here and in the revised manuscript - changes highlighted in green), both theoretically and empirically. We hope these changes shed lights on the remaining doubts, and that the reviewer will consider increasing the score according to the substantial extensions made to the submission.
>
> **Weaknesses**
>
> **On differentiability of the projection layer:**
>
> - We thank the reviewer for their insightful comment. To start, we stress that given that, in most cases, the layers of the backbone network will be at most *almost* everywhere differentiable (e.g., for ReLU activations), all one may be interested in is to have almost everywhere differentiability of the projection layer. In fact, our experiments show that in practice this is sufficient for markedly surpassing the state of the art.
> - For this, we have strong theoretical insights on the soundness of our layer. Note that, on Euclidean spaces (indeed, on Hilbert spaces) the projection operator is globally 1-Lipschitz (see, e.g., Proposition 4.16  in [1], and recall that firmly non-expansive implies 1-Lipschitz). As such, by invoking Rademacher’s theorem [2, Theorem 1.4] we can conclude that the projection layer is almost everywhere differentiable (in the Lebesgue measure sense).
> - Regarding the application of the Implicit Function Theorem (IFT), under local differentiability conditions (such as the ones  proposed by the reviewer) our IFT formula is correct. The way that related works handle non-smooth points is to either (i) show that if the linear system is non-singular, then multiple solutions exist and they belong to the subdifferential (see [3, Appendix C.1]) or (ii) they resort to heuristics, e.g., by solving a least-squares approximation rather than the linear system itself (see [4, Section 4.3]. We similarly employ a heuristic by running finitely many BiCGSTAB iterations, which is very computationally attractive as it avoids run-time checks and, in our numerical experience, is quite stable. Please see also our ablation results in the newly added Appendix F.3 (attached below), in which we compare to finite differences.
> - Interestingly, we can define an IFT even on non-smooth points by employing a recent generalization of the subdifferential, namely *conservative gradients* [5], that is valid for non-smooth locally Lipschitz, mappings. We refer to the seminal works [5], [6] for a non-smooth variant of IFT that can be employed for backpropagation of NN models and machine learning applications. In this framework, the IFT and the chain rule used for backpropagation are true under very mild conditions. Roughly speaking, this would correspond to our operator $\Phi(x, y)$being semialgebraic and contractive with respect to $y$ [7]. Then, we could conclude that our IFT rule is correct everywhere. We discuss the connections to non-smooth IFT in the revised manuscript to show the background math.
> - Lines 161-163: Please see our detailed response above.
> - Lines 173-179: Note that, we design the sets $\mathcal{A}$ and $\mathcal{K}$ , such that $\mathcal{C} = \Pi_d(\mathcal{A} \cap \mathcal{K})$, where $\Pi_d$ denotes the selection of the first $d$ coordinates of the set. Under this choice, the optimization problems (1) and (2) have the same unique solution, due to standard properties of the indicator function. In lines 145-161 and Appendix E, we show how $\mathcal{A}$ and $\mathcal{K}$ can be designed to satisfy our specification for various $\mathcal{C}$, namely general polytopes, second-order cone constraints, simplices, and intersections and Cartesian products of such sets. Please let us know if this answers your question.
> - Eqs (3a-3c): Our strict feasibility assumption (formally Assumption 1, line 1636) is derived from Corollaries 27.6 and 28.3 in [1] (a standard operator theory textbook). To the best of our knowledge, non-empty intersection suffices only when $\mathcal{A}$ and $\mathcal{K}$ are polyhedral. We would be grateful if the reviewer could point us to a reference discussing the sufficiency of the proposed assumption.
> - Lines 218-228: Please see our response above.
> - **Recommendations**:
>     - We would like to highlight that we do not claim "guaranteed differentiability" anywhere on our paper. Nonetheless, we will clarify the empirical nature of our results by including a summary of our previous discussion on differentiability.
>     - We will include a citation and discussion on the IFT validity, as outlined in our previous discussion.
> - **Questions:**
>     - We thank the reviewer for their question. Indeed, OT problems are a potential application for hard-constrained neural networks. We have not investigated how $\Pi$net compares to OT solvers, but it is an interesting direction for future work.
>
> Please see the next comment for references and attachments.

---

> ### Author Response · Authors · 2025-11-21
> **Response of Authors to Reviewer yrzg - References and Attachments**
>
> **Attachments**
> https://imgur.com/a/DCX1Aqk
>
> **References**
>
> [1] H.H. Bauschke and P.L. Combettes. Convex Analysis and Monotone Operator Theory in Hilbert
> Spaces. 2017.
>
> [2] Simon, Leon. "Lectures on geometric measure theory." 1984.
>
> [3] Amos, Brandon, and J. Zico Kolter. "Optnet: Differentiable optimization as a layer in neural networks."
>
> [4] Agrawal, Akshay, et al. "Differentiable convex optimization layers."
>
> [5] Bolte, Jérôme, and Edouard Pauwels. "Conservative set valued fields, automatic differentiation, stochastic gradient methods and deep learning."
>
> [6] Bolte, Jérôme, et al. "Nonsmooth implicit differentiation for machine-learning and optimization."
>
> [7] Bolte, Jérôme, et al. "Differentiating nonsmooth solutions to parametric monotone inclusion problems."

---

### Official Review · Reviewer_hExW · 2025-10-31

**Soundness:** 3
**Presentation:** 3
**Contribution:** 3
**Rating:** 8
**Confidence:** 4

**Summary:**

The paper introduces a differentiable projection network (ΠNet) for handling hard constraints within end-to-end neural network training. The key idea is to embed a projection operator directly as a network layer, ensuring that outputs always satisfy predefined linear equality and inequality constraints. The projection is computed via a Douglas–Rachford splitting scheme, which alternates between projections onto an affine set and a box set until convergence to their intersection. For backpropagation, ΠNet avoids differentiating through the iterative steps. Instead, it applies the Implicit Function Theorem (IFT) at the fixed point of the DR operator, computing the vector–Jacobian product by solving a linear system. Overall, the method provides a principled, computationally efficient, and differentiable mechanism for enforcing hard linear constraints in neural networks.

**Strengths:**

The paper is well organized and clearly written, with consistent notation and a coherent presentation.

It introduces a conceptually clean and mathematically principled approach to making projection-based constraint enforcement differentiable. By combining the Douglas–Rachford splitting scheme with the Implicit Function Theorem at the fixed point, the authors design a projection layer that is both efficient and end-to-end trainable. This integration of classical operator-splitting methods with modern differentiable programming is novel and results in a solver-free alternative to existing differentiable optimization layers such as cvxpylayers, DC3, and JAXopt.

The technical development is sound and internally consistent. The backward pass through the fixed-point system is derived correctly, and the implicit differentiation is implemented carefully via the solution of a linear system. The experimental evaluation is thorough: benchmarks include both convex and nonconvex problems across multiple scales, and comparisons are made against strong baselines. ΠNet achieves comparable or superior constraint satisfaction while reducing computation time by one to two orders of magnitude, demonstrating clear numerical advantages.

ΠNet provides a practical and general framework for incorporating hard constraints into neural networks without compromising differentiability. Its efficiency and stability make it particularly relevant for constrained learning tasks in control, trajectory optimization, and physics-informed modeling, representing a meaningful contribution to the literature.

**Weaknesses:**

While the paper is technically solid, several aspects could be clarified or strengthened.

First, the claim of handling convex constraint sets is somewhat overstated. All derivations and experiments are limited to linear (affine) equality and inequality constraints. Although these are convex in the geometric sense, the current method does not address more general convex sets such as norm balls, SOCs, or PSD cones. The projections and corresponding Jacobians for these sets are nontrivial, and it remains unclear whether the Douglas–Rachford formulation or the implicit differentiation strategy would remain computationally tractable in those cases. Clarifying this limitation and outlining potential extensions would make the contribution more precise and complete.

In addition, the experimental comparisons, though comprehensive, could be made fairer and more reproducible. JAXopt and DC3 are trained under different budgets and tolerances. Reporting results under equal wall-clock time or identical training epochs would offer a more balanced assessment.

Finally, although the backward pass based on the Implicit Function Theorem is elegant, the paper does not discuss the conditions ensuring the stability or invertibility of the associated linear system. A short clarification or empirical remark on this aspect would strengthen the completeness of the work.

**Questions:**

1. The choice of $\omega = 1.7$ is mentioned as default but not justified. Did the authors observe sensitivity in convergence or gradient stability with different values of $\omega$? Providing even brief empirical guidance would be useful for practitioners.
2. The method uses a fixed number of Douglas–Rachford iterations at inference equal to that during training. Have the authors verified that performance remains stable if more (or fewer) iterations or early-stopping are used at test time? This could reveal whether ΠNet truly learns a stable fixed-point operator or overfits to the training iteration budget.

---

> ### Author Response · Authors · 2025-11-21
> **Response from Authors to Reviewer hExW - Weaknesses**
>
> We thank the reviewer for their very positive evaluation. We believe to have addressed all the comments (here and in the revised manuscript - changes highlighted in blue). We hope these changes highlights the strength of our method, and that the reviewer will consider increasing the score accordingly.
>
> **Weaknesses**
>
> **General convex sets**: Although it is true that in the main part of our paper our exposition and experiments are limited to affine constraints, we would like to stress that we presented derivations for non-linear constraints in Appendix E (including $\ell_1-\text{balls}$, SOCs, and simplices) and short experiments for SOCs in Appendix B.5. Following the reviewer’s suggestion (also from rev. o3mf) we have significantly extended our evaluation on SOCs, and performed the same extensive benchmarks as for affine constraints. Our results showcase the applicability of our framework to general convex sets. Please refer to our response to rev. o3mf - Weakness 1 for more details. We hope that these additional experiments answer the reviewer’s comment. Further, note that for any convex set consisting of intersections and products of all the sets listed in the reply to rev. o3mf our framework is computationally efficient, and the projection and Jacobians can be computed in closed-form. Given the comments from multiple reviewers, we acknowledge our exposition possibly does not emphasize enough this strength of our approach, and we attempted to do so in the revised manuscript. The changes related to this are in orange (rev. o3mf).
>
> **Fairness of benchmarks:** We thank the reviewer for giving us the opportunity to highlight that the benchmarks are unfair towards our proposed method. To be concrete:
>
> - **Training budget:** From the learning curves (Figures 3, 6, 7, and 8), we see that $\Pi$net is always given (significantly) less wall-clock time for training. We determined the wall-clock time of $\Pi$net and JAXopt such that satisfactory performance is attained, and for DC3 we used the default. Yet, it consistently outperforms other methods in terms of solution quality (Figures 2 and 5).
> - **Tolerances:** Compared to JAXopt, $\Pi$net satisfies lower tolerance for CV and attains faster inference time. Therefore, using the same tolerance would only widen the inference time gap (in favor of $\Pi$net). Compared to DC3, for small problems both methods are within the CV threshold that we set. For large problems, DC3 exhibits substantial constraint violation whereas $\Pi$net does not. To reduce CV by DC3, we tried tuning the number of correction steps (i.e., increasing the default value), as shown in Figure 9, but unfortunately this did not help much.
>
> On a side note, notice that the entire wall-clock budget of $\Pi$net is less than a single epoch of JAXopt (on any of the benchmark problems).
>
> **On differentiability of backward pass:** We thank the reviewer for highlighting this important aspect of our work. We have included an extensive theoretical discussion on the differentiability of our projection layer (in Appendix F of our revised manuscript), showing that it is indeed almost everywhere differentiable and also pointing to promising theoretical frameworks that would allow one to use the implicit function even on non-smooth instances. Further, we provide empirical evidence of the performance and accuracy of our backward pass compared to a finite differences "ground truth". We refer to our response to rev. yrzg for details and additional discussions on invertibility of the matrix.

---

> ### Author Response · Authors · 2025-11-21
> **Response from Authors to Reviewer hExW - Questions**
>
> **Questions:**
>
> 1. Choosing $\omega$ close to 1.5 is common among successful modern implementations of DR or ADMM. For example, OSQP [1] uses 1.6, SCS [2] uses 1.5, and other works indicate similar choices [3], [4]. Accordingly, we found that $\omega = 1.7$, or similar values, improved the convergence rate of our projection scheme (sometimes by a lot). See also our ablation results on $\omega$ in the response to reviewer 1VJT.
> 2. Thank you for the insightful question. We have extensively tested this, with both more and fewer iterations at test time. Using more iterations works well and stably, and is meaningful to achieve reduced constraint violation. Using less iterations needs more care. Slightly less iterations is fine and can improve inference time. Significantly less might cause issues since the iterates might not be close to a fixed-point anymore. Of course, "slightly" and "significantly" here depend on the problem at hand. We have performed some additional ablations to show exactly this behaviour. In the following image, we show the RS and CV obtained by training $\Pi$net with 50 iterations, and then deploying with different number of iterations during testing (we write, e.g., test = 50, in the legend for 50 iterations at testing time). See image below. We consider such insights to be critical for practitioners and we plan on compiling a list observations and guidelines in the documentation of our code. We added this ablation and discussion to the revised manuscript, available also at the link below. Perhaps interestingly, note that the RS is invariant to the number of iterations during testing, whereas the CV gradually decreases. This is true as long as we have enough iterations: as the number of iterations during test becomes too small (e.g., 2) also the RS is affected: The output is not close to the projection anymore.
>
> https://imgur.com/a/NKEHhLa
>
> [1] Stellato, Bartolomeo, et al. "OSQP: An operator splitting solver for quadratic programs."
>
> [2] O’donoghue, Brendan, et al. "Conic optimization via operator splitting and homogeneous self-dual embedding."
>
> [3] Eckstein, Jonathan. "Parallel alternating direction multiplier decomposition of convex programs."
>
> [4] Eckstein, Jonathan, and Michael C. Ferris. "Operator-splitting methods for monotone affine variational inequalities, with a parallel application to optimal control."

---

### Official Review · Reviewer_o3mf · 2025-10-31

**Soundness:** 3
**Presentation:** 3
**Contribution:** 2
**Rating:** 6
**Confidence:** 4

**Summary:**

This paper proposes PiNet, a differentiable layer for solving projection problems over convex constraints to ensure neural network output feasibility. The key contribution is a decomposition approach for certain constraints that can be represented as the intersection of hyperplanes and boxes, where each component admits efficient projection. The method applies operator splitting algorithms to the projection problem in this decomposed form, with the backward pass implemented via implicit gradients. Experiments demonstrate improved efficiency and reduced optimality gaps compared to baseline methods on benchmark problems and multi-vehicle motion planning tasks.

**Strengths:**

1. Ensuring neural network output feasibility is important for real-world applications, and this work addresses a practical and relevant problem.
2. The combination of constraint decomposition and operator splitting algorithms appears to be novel, and significantly improves projection efficiency for certain types of constraints.

**Weaknesses:**

1. The range of constraints that admit the decomposition form of hyperplanes and boxes is unclear. Besides the examples provided in the appendix, can the authors provide a clear definition or characterization of the types of constraints that fall within this framework?
2. The experiments mainly focus on linear constraints. Evaluation of non-linear constraints would further strengthen the contribution and demonstrate the broader applicability of this work.
3. The authors mention that Min et al. (2024) propose a closed-form expression to recover feasibility for polyhedral constraints. It would be valuable to include a comparison with this recent work to better position the proposed approach.
4. The related work section on hard-constrained neural networks could be strengthened by including other approaches, such as reparameterization approaches [1-2] and sampling-based approaches [3].

- [1] Tabas D, Zhang B. Safe and efficient model predictive control using neural networks: An interior point approach. CDC 2022.
- [2] Liang E, Chen M, Low SH. Homeomorphic projection to ensure neural-network solution feasibility for constrained optimization. JMLR 2024.
- [3] Kratsios A, Zamanlooy B, Liu T, Dokmanić I. Universal approximation under constraints is possible with transformers. ICLR 2022.

**Questions:**

Please refer to the weakness section.

Overall, this work is well-written. I will adjust my rating upon clarification of the above points and possible additional experiments addressing the weaknesses mentioned.

---

> ### Author Response · Authors · 2025-11-21
> **Response from Authors to Reviewer o3mf - Weakness 1**
>
> We thank the reviewer for the time spent in crafting this review. We believe to have addressed all the comments (here and in the revised manuscript - changes highlighted in orange). We hope these changes highlight the strength of our method, and that the reviewer will consider increasing the score accordingly.
>
> **Weaknesses**
> 1. We thank the reviewer for their clarifying question. Fundamentally, we can handle any convex set $\mathcal{C}$ that can be written as $\mathcal{C} = \Pi_d(\mathcal{A} \cap \mathcal{K})$, for which $\Pi_\mathcal{A}$ and $\Pi_\mathcal{K}$ are computable. Next, we specify one very common family of sets that admits such a decomposition:
> $$ A_\text{eq} y = b_\text{eq} \quad (\text{Eq}) $$
> $$ \ell_\text{box} \leq y \leq u_\text{box} \quad (\text{Box}) $$
> $$ \ell_\text{ineq} \leq A_\text{ineq} y \leq u_\text{ineq} \quad (\text{Ineq}) $$
> $$ \lVert A_{\text{soc}, i} y + a_{\text{soc,} i} \rVert_2 \leq f_{\text{soc}, i}^{\top} y + b_{\text{soc}, i}, \quad i = 1, \ldots, N_\text{soc} \quad (\text{SOC}) $$
> $$\lVert A_{\ell_1, i} y + a_{\ell_1, i} \rVert_1 \leq f_{\ell_1, i}^{\top} y + b_{\ell_1, i}, \quad i = 1, \ldots, N_{\ell_1} \quad (\ell_1-\text{ball}) $$
> $$\lVert A_{\ell_\infty, i} y + a_{\ell_\infty, i} \rVert_\infty \leq f_{\ell_\infty, i}^{\top} y + b_{\ell_\infty, i}, \quad i = 1, \ldots, N_{\ell_\infty} \quad (\ell_\infty-\text{ball}) $$
> $$ y_{\mathcal{S}j} \geq 0, \quad \lVert y_{\mathcal{S}j} \rVert_1 = 1, \quad \text{for index sets }\mathcal{S}_j \quad (\text{Simplex}) $$
>
> where $y$ is the decision variable, and we used matrices and vectors of compatible dimensions. Any set of this form can be decomposed in the way we present, see Appendix E "The intersection of previous sets" for an example. We highlight this family of sets because (i) it is ubiquitous in practice, and (ii) our method is very efficient on it since we can choose $\mathcal{A}$ and $\mathcal{K}$ with $\Pi_\mathcal{A}$ and $\Pi_\mathcal{K}$ having closed-form expression. Some additional remarks:
>
> - Note that, the sets $(\text{Box})$, $(\ell_1-\text{ball})$, $(\ell_\infty-\text{ball})$ and $(\text{Simplex})$ can be written also as $(\text{Ineq})$. We write them down separately to indicate that we can explicitly handle them more efficiently than general affine inequality constraints.
> - Given a set $\mathcal{C}$ described as above, one can automatically compute its decomposition onto $\mathcal{A}$ and $\mathcal{K}$, hence making the approach user-friendly. Our software already provides this automatic decomposition functionality for $\mathcal{C}$ consisting of intersections of $(\text{Box}),$ $(\text{Eq})$ and $(\text{Ineq})$. We plan on extending this for all of the above elementary sets.
>
> We revised the manuscript to clarify that while $\Pi$net can handle any convex set with computable Euclidean decomposition, our focus is on the sets listed in the response as those are the ones that are most interesting in practice.

---

> ### Author Response · Authors · 2025-11-21
> **Response from Authors to Reviewer o3mf - Weakness 2**
>
> **Weaknesses**
>
> 2. We thank the reviewer for their suggestion, indeed in Appendix B.5 we already present $\text{SOC}$ constraints, but we do not thoroughly benchmark it.  In our revised manuscript, we added to Appendix B.5 extensive benchmark results also for non-linear constraints. Specifically, we consider as constraints an intersection of $(\text{Eq})$ and $(\text{SOC})$, and note that the latter is non-linear. The exact formulation is given in Appendix B.5. We compare against:
>
> a. $\texttt{cvxpylayers}$: a differentiable optimization solver which we use as a replacement of our custom projection layer. (This is the same procedure we adopted to benchmark our method against $\texttt{JAXopt}$ for linear constraints; $\texttt{JAXopt}$ does not support SOC constraints.)
>
> b. $\texttt{SCS}$: a traditional first-order solver for SOC programs.
>
> As with our linearly-constrained benchmarks, we consider learning curves, RS vs CV, and inference times. Our results indicate that:
>
> a. Compared to $\texttt{cvxpylayers}$, $\Pi$net can train and perform inference significantly faster, even orders of magnitude for large problems. $\Pi$net provided solutions with both lower RS and CV.
>
> b. Compared to $\texttt{SCS}$, $\Pi$net is faster at inference for a batch of small ($d_1 = d_2 = 25$) problems, and both a single and a batch of large ($d_1 = d_2 = 500$) problems. Instead, $\texttt{SCS}$ is faster for single small problems.
> **Note:** To make our comparison with $\texttt{SCS}$ as fair and comprehensive as possible, we performed two benchmarks. On one, we interfaced $\texttt{SCS}$ through the very popular parser $\texttt{CVXPY}$ using its Disciplined Parametrized Programming functionality, as many users would do. On the other benchmark, we interfaced $\texttt{SCS}$ directly and exploited its native parametric programming functionality, as more advanced users would. We would like to highlight that with these benchmarks we are exploiting the full parametric capabilities of existing solvers, which is not always the case by end-users or even other HCNN benchmarks.
>
> **Attachments**: We anonymously posted extracts from the revised manuscript for your convenience at https://imgur.com/a/WWni5jZ

---

> > ### Author Response · Authors · 2025-11-21
> > **Response from Authors to Reviewer o3mf - Weaknesses 3 and 4**
> >
> > 3. We are working on the code from Min et al. (2024) and will attempt to obtain the results from their method for the requested benchmarks within the review time. We prioritized the benchmarks on the SOC constraints as we believe them to bring more insights on the advantages of our method.
> >
> > 4. We thank the reviewer for pointing out these related works. We have included them in our revised manuscript. Specifically, [1] is a great example of application of HCNN to MPC (we added a reference there), [2] further shows how convex constraints are not in fact limiting, and projection on balls is already very general (we added the reference in our limitation section), [3] shows universal approximation properties and employ a probabilistic sampling approach, and we now discuss it in the introduction.

---

### Author Response · Authors · 2025-12-04
**Closing Remarks**

In these closing remarks, we would like to thank the reviewers for their very positive feedback that highlighted:

- the significance of our work (Rev. o3mf: “ this work addresses a practical and relevant problem”, Rev. hExW: “[…] representing a meaningful contribution to the literature”, “The experimental evaluation is thorough”, Rev. yrzg: “А mathematically sound appendable layer with a robust Forward pass formulation”, Rev. 1VJT: “The layer can be seamlessly attached to any backbone network”).
- its clarity and quality (Rev. hExW: “well organized and clearly written”, Rev. o3mf: ”Overall, this work is well-written“);

We acknowledge the insightful comments of the reviewers that gave us the opportunity to further improve our work and to better highlight its advantages. Particularly, we briefly summarize (some of) the substantial content added to our paper during the rebuttal process, and we refer to our revised manuscript for more details:

- We included a clear formulation for a general and very practically-relevant family of constraints that are amenable to our projection layer.
- We performed a whole new set of benchmarks on non-linear constraints. Specifically, we considered second-order cone constraints and compared $\Pi$net to both a learning-based approach ($\texttt{cvxpylayers}$) and a traditional solver ($\texttt{SCS})$. We showed not only that $\Pi$net can seamlessly handle non-linear constraints, but also that it can outperform the considered baselines, in certain cases by a lot.
- We included many new ablations on various hyperparameters, such $\sigma$, $\omega$, number of backwards iterations, and number of test iterations (vs training iterations). With these, we demonstrated that $\Pi$net, and specifically our custom projection layer, is not sensitive to hyperparameter tuning.
- We clarified that, in our original benchmarks our evaluation was unfair towards our proposed method, $\Pi$net, but nonetheless it was able to outperform competing approaches. Specifically, we allowed $\Pi$net less computational budget (wall-clock time) for training. Also, $\Pi$net in most cases demonstrated less constraint violation during inference, without compromising inference time.
- We included a technical discussion to strengthen the theoretical soundness of our projection layer. Particularly, we showed that our layer is almost everywhere differentiable (in the Lebesgue measure sense), as is the case for most NN layers. Further, we discussed how our backpropagation rule based on the Implicit Function Theorem could potentially even be extended to non-smooth points. This is a very interesting research avenue for future theory-focused papers. For this, we referred to recent works on non-smooth calculus and proposed a clear way to attack this problem.
- We added suggested relevant works.

Through all these additions, we believe to have substantially answered the reviewers’ questions and further clarified the strengths of our approach.

---

### Meta-Review · Area_Chair_2gnu · 2026-01-07

**Summary:**

Reviewers praised:
- The clean and principled approach to making projection-based constraint enforcement differentiable.
- Soundness and consistency of the approach.
- Practicality and generality of PiNet.

Reviewers were concerned about:
- Some overstated claims in the paper with respect to handled constraints.
- Fairness and reproducibility of the experimental comparisons.
- Conditions for stability/invertibility of the associated linear system (re: the backward pass).
- Differentiability of the projection layer.

**Reviewer Concerns:**

Addressed:
- Clarifications on the claims in the paper with respect to handled constraints.
- Conditions for stability/invertibility of the associated linear system (re: the backward pass).
- Fairness and reproducibility of the experimental comparisons.
- Differentiability of the projection layer.

**Reviewer Scores:**

All reviewers are generally positive on the paper. The paper focuses on a foundational problem and proposes an elegant solution.

It is theoretically possible that o3mf would have raised their score to a 8, but not guaranteed.

---

### Decision · Program_Chairs · 2026-01-26

Accept (Oral)